# CAPTAIN CINEMA: TOWARDS SHORT MOVIE GENERATION

**Junfei Xiao**[1,2]  **Ceyuan Yang**[2*]  **Lvmin Zhang**[3]  **Shengqu Cai**[2,3]  **Yang Zhao**[2]
**Yuwei Guo**[2,4]  **Gordon Wetzstein**[3]  **Maneesh Agrawala**[3]  **Alan Yuille**[1]  **Lu Jiang**[2]
[1]Johns Hopkins University  [2]ByteDance Seed  [3]Stanford University  [4]CUHK

## ABSTRACT

We present **Captain Cinema**, a generation framework for short movie generation. Given a detailed textual description of a movie storyline, our approach firstly generates a sequence of keyframes that outline the entire narrative, which ensures long-range coherence in both the storyline and visual appearance (e.g., scenes and characters). We refer to this step as top-down keyframe planning. These keyframes then serve as conditioning signals for a video synthesis model, which supports long context learning, to produce the spatio-temporal dynamics between them. This step is referred to as bottom-up video synthesis. To support stable and efficient generation of multi-scene long narrative cinematic works, we introduce an interleaved training strategy for Multimodal Diffusion Transformers (MM-DiT), specifically adapted for long-context video data. Our model is trained on a curated cinematic dataset consisting of interleaved samples for video generation. Our experiments demonstrate that Captain Cinema performs favorably in the automated creation of visually coherent and narratively consistent short films.

## 1 INTRODUCTION

Narrative is central to how humans communicate, remember, and perceive the world. As Bruner has argued (Bruner, 1991), narrative structures are fundamental to organizing human experience, while Harari (Harari, 2014) emphasizes that shared stories underpin the development of societies. In the field of video generation (Ho et al., 2022b; Singer et al., 2022; Chen et al., 2023; 2024b; Ho et al., 2022a; Wang et al., 2023; Zhou et al., 2022), recent advances, especially in diffusion-based (Peebles & Xie, 2023; Brooks et al., 2024; Hong et al., 2022; Yang et al., 2024c) and auto-regressive models (Kondratyuk et al., 2024; Sun et al., 2023b; 2024; Wang et al., 2024b) have enabled impressive progress in synthesizing short video clips for tasks such as image animation (Chen et al., 2025; Xu et al., 2024), video editing (Feng et al., 2024; Ceylan et al., 2023), and stylization (Huang et al., 2024b). However, these accomplishments predominantly address visual fidelity and local temporal coherence, leaving the deeper challenge of producing videos that convey coherent, engaging narratives over extended durations. Bridging this gap—from visually plausible snippets to full-length, story-driven videos—constitutes a pivotal frontier for research and practice.

Automating the generation of feature-length cinematic narratives remains underexplored. Achieving narrative coherence and visual consistency over extended durations demands models that capture long-range dependencies while retaining fine-grained detail. Existing approaches frequently encounter exploding context lengths, and visual drift when scaled to longer videos. To overcome these challenges, we propose **Captain Cinema**, a framework tailored for story-driven movie synthesis.

`Captain Cinema` balances global plot structure with local visual fidelity through two complementary modules. A *top-down* planner first produces a sequence of key narrative frames that outline the storyboard, ensuring coherent high-level guidance. A *bottom-up* video synthesizer then interpolates full motion conditioned on these keyframes, maintaining both narrative flow and visual detail. Central to this design is `GoldenMem`, a memory mechanism that selectively retains and compresses contextual information from past keyframes. By summarizing long histories without exceeding memory budgets, `GoldenMem` preserves character and scene consistency across multiple scenes with super-long context, enabling scalable generation of multi-scene videos.

---

*Corresponding author.

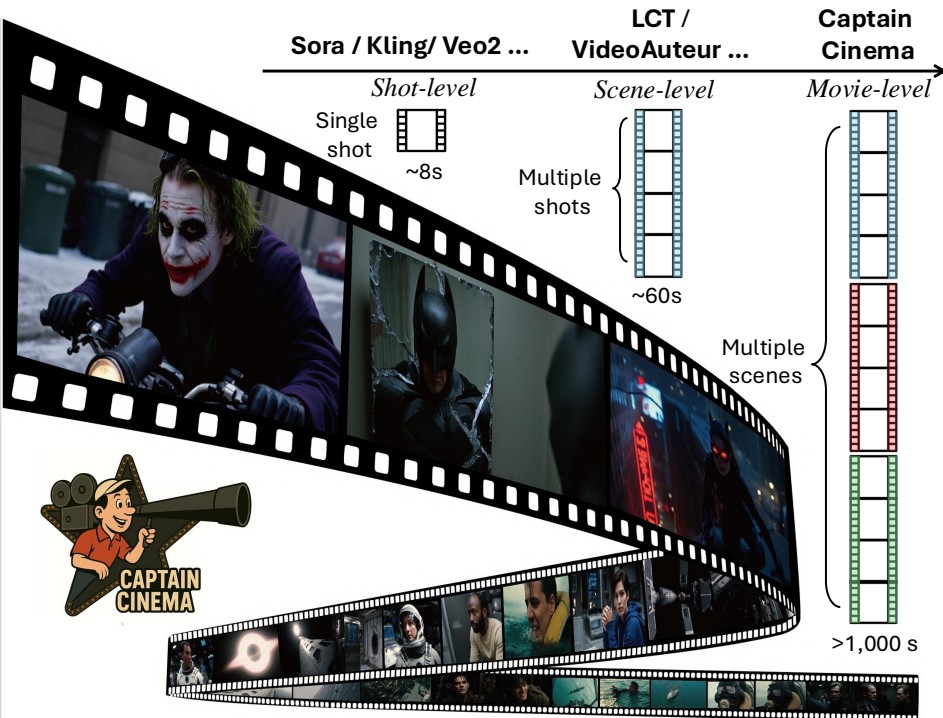

Figure 1: **Captain Cinema: "I can film this all day!"** Captain Cinema bridges top-down interleaved keyframe planning with bottom-up interleaved-conditioning video generation, taking a step toward the first multi-scene, whole-movie generation, preserving superior visual consistency in scenes and identities across the long video. All the movie frames are **generated**.

Additionally, we build a specialized data processing pipeline for processing long video data for movie generation and introduce progressive long-context tuning strategies tailored for Multimodal Diffusion Transformers (MM-DiT). These techniques enable stable and efficient fine-tuning on large-scale, long-form cinematic datasets, addressing the challenges of multi-scene video generation. Extensive experiments and ablation studies demonstrate that `Captain Cinema` not only achieves strong performance in long-form narrative video synthesis but also enables the automated creation of visually consistent short films that significantly exceed the duration of existing works, setting a new milestone in multimodal video generation capabilities.

## 2 RELATED WORKS

**Text-to-Video Generation.** Text-to-video models (Ho et al., 2022b; Singer et al., 2022; Chen et al., 2023; 2024b; Ho et al., 2022a; Wang et al., 2023; Zhou et al., 2022) now generate 5–10 s clips with high visual fidelity. Most adopt a latent diffusion paradigm (Rombach et al., 2022); diffusion variants such as DiT (Peebles & Xie, 2023), Sora (Brooks et al., 2024), and CogVideo (Hong et al., 2022; Yang et al., 2024c) extend this design with larger datasets and refined denoisers. Autoregressive alternatives like VideoPoet (Kondratyuk et al., 2024) and the Emu family (Sun et al., 2023b; 2024; Wang et al., 2024b) instead predict discrete visual tokens sequentially. These approaches remain clip-centric and lack mechanisms for narrative or visual consistency over longer horizons. We bridge this gap by introducing explicit consistency regularisation for full-length movie generation.

**Interleaved Image–Text Modeling & Conditioning.** Interleaved image–text generation (Dong et al., 2023; Tian et al., 2024a; Ge et al., 2024; Yang et al., 2024a; Alayrac et al., 2022) has emerged as a promising paradigm -for producing richly grounded multimodal content. Early efforts (Radford et al., 2021; Li et al., 2023b; Sun et al., 2023a) leveraged large-scale image–text corpora (Gadre et al., 2024; Schuhmann et al., 2022) but were largely restricted to single-turn tasks such as captioning or text-to-image synthesis. The advent of large language models (Touvron et al., 2023) and unified vision–language architectures (Li et al., 2023a; Liu et al., 2024; Wang et al., 2024a) has

enabled more sophisticated interleaved reasoning, yet most existing systems generate content only once and overlook coherence across multiple shots. VideoAuteur (Xiao et al., 2024) addresses this problem through an interleaved VLM director and LCT (Guo et al., 2025) directly finetunes MM-DiTs to peform long interleaved video generation. Although Diffusion Forcing (Chen et al., 2024a), LTX-Video (HaCohen et al., 2024a) and Pusa (Liu & Liu, 2025) support multi-keyframe conditioning, they remain confined to single-shot scenarios. By contrast, our approach addresses multi-shot interleaved conditioning, explicitly modeling cross-shot coherence for long-form video generation.

**Narrative Visual Generation.** Existing research on narrative visual generation primarily focuses on ensuring semantic alignment and visual consistency. Recent methods—VideoDirectorGPT (Lin et al., 2024), Vlogger (Zhuang et al., 2024), Animate-a-Story (He et al., 2023), VideoTetris (Tian et al., 2024b), IC-LoRA (Huang et al., 2024a), and StoryDiffusion (Zhou et al., 2024)—pursue these goals through diverse architectural and training strategies. While most prior work concentrates on producing semantically consistent image sets (Huang et al., 2024a; Yang et al., 2024b; Zhou et al., 2024), our objective is to generate fully coherent narrative videos. Although certain approaches condition synthesis on text (Zhuang et al., 2024; Xu et al., 2024) or augment prompts with keyframes (Zhao et al., 2024), our framework explicitly disentangles long-form generation into (i) interleaved keyframe synthesis and (ii) multi-frame-conditioned video completion. This design achieves state-of-the-art results in long narrative movie generation, offering superior visual fidelity, temporal coherence, and robustness to super-long context conditions.

**Visual Generation with Token Compression.** The computational demands associated with synthesizing high-resolution visual content for video sequences necessitate efficient data representation strategies. FramePack (Zhang & Agrawala, 2025) employs context packing in video generation to compress input frames, thereby maintaining a fixed context length for enhanced processing efficiency. MoC (Cai et al., 2026) instead trains a learnable dynamic sparse attention router to select relevant contexts at the attention level. FlexTok (Bachmann et al., 2025) utilizes an adaptive 1D tokenizer for images, adjusting token sequence length according to image complexity to achieve semantic compression. LTXVideo (HaCohen et al., 2024b) uses compressed VAEs to improve model efficiency. PyramidFlow (Jin et al., 2024) performs video diffusion in a pyramid and renoises the multi-level latents. FAR (Gu et al., 2025) proposes a multi-level causal history structure to establish long-short-term causal memory and discuss KV cache integrations. HiTVideo (Zhou et al., 2025) uses hierarchical tokenizers in video generation models and connect to autoregressive language models. Memory in discrete space like keyframes is less discussed but plays an important role in our framework to process frame contexts, optimized for synthesizing long-form narrative visuals.

## 3 METHOD

### 3.1 LEARNING LONG-RANGE CONTEXT IN MOVIES

Different from (Guo et al., 2025; Huang et al., 2024a) trained on single-scene data, we directly learn the movie generation task from entire movies. Specifically, our keyframe generation model is trained on interleaved image-text pairs, while our video generation model learns from interleaved video-text pairs, significantly enhancing its generalization across scenes and cinematic contexts.

**Data.** We collect data from public available sources as our experimental dataset, amounting to a total length of approximately 500 hours. We process whole movies with the following data pipeline to produce interleaved keyframe-text pairs as well as shot-text pairs. After processing these data, we obtained roughly 300,000 keyframes and video shots to train our top-down interleaved keyframe generation model (§3.2) and the bottom-up interleaved conditioned video generation model (§3.3).

**Processing & Filtering.** We detect scene cuts with PySceneDetect (Breakthrough, 2021) and extract a mid-clip frame from each resulting segment. After removing black borders, we center-crop to a 2:1 aspect ratio and resize so the shorter side is 400 pixels. We then run Gemini-2.0 Flash (Gemini Team, 2023) to discard low-quality or uninformative frames and to generate detailed captions for the remaining keyframes. To annotate characters, we include consistent `<character name>` tags to preserve IPs. Section B details the prompt used for keyframe annotation.

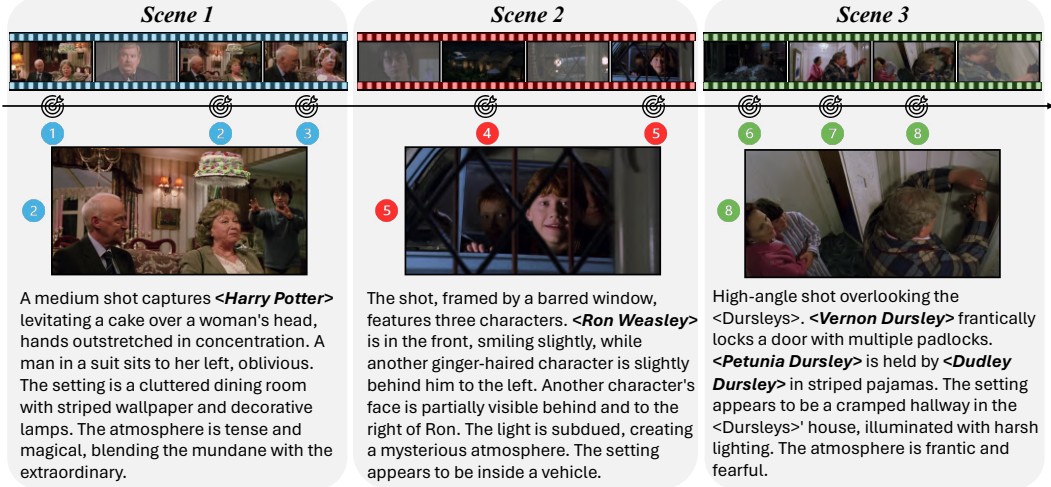

Figure 2: **Learn from whole movies.** Here is an interleaved data sample processed from full-length movies. Our data pipeline extract structured narrative and visual information across scenes. Each frame is annotated with detailed visual description with major <Character Names> of movies.

## 3.2 TOP-DOWN INTERLEAVED KEYFRAME PLANNING

In this section, we introduce our method to finetune a pretrained Text-to-Image Model (i.e., Flux 1.Dev (Labs, 2024)) for interleaved keyframe generation with stability and efficiency.

**Hybrid Attention Masking with MM-DiT.** Our design builds on Flux, whose model is split into $D$ *double-stream* (image–text) blocks followed by $S$ *single-stream* blocks. Given $P$ image–text pairs $S = \{(x_i, y_i)\}_{i=1}^P$, with $x_i \in \mathbb{R}^{L_I \times d}$ and $y_i \in \mathbb{R}^{L_T \times d}$, we concatenate the tokens as $z_i = [x_i \| y_i]$.

1. *Local (double-stream) blocks.* Each of the first $D$ blocks uses a block–diagonal mask $M_{\text{local}} = \text{diag}(M_1, \ldots, M_P)$; such that $z_i$ attends only to itself.

2. *Global (single-stream) blocks.* Their outputs are concatenated $Z = [z_1 \| \ldots \| z_P]$ and passed to the next $S$ blocks. A full mask $M_{\text{global}} \equiv 0$ enables bi-directional inter-pair attention during training; swapping it for an upper-triangular mask yields causal, auto-regressive generation.

The two types of masking (cf., Fig. 3b) keeps early computation local and efficient while later blocks aggregate global context, achieving coherent interleaved key-frame generation.

**GoldenMem: Compress Long-Context Visual Memory**. As we target movie generation, how to design a long-context memory bank of generated visual frames is a challenge when the context growth. Inspired by the golden ratio squares with an upper bound area, we propose GoldenMem to compress long context visual frames through golden ratio downsamling with semantic-oriented context selection. To maintain a long visual context without inflating the token budget, we store only the current frame at full resolution and encode earlier frames at progressively coarser scales. Let the golden ratio be $\varphi = (1 + \sqrt{5})/2 \approx 1.618$ and denote the short side of the newest latent by $s_0$ (e.g., $s_0 = 25$). We downsample the $i$-th previous frame by

$$s_i = \lfloor s_{i-1}/\varphi \rfloor, \qquad i = 1, \ldots, k,$$

which yields the inverse-Fibonacci sequence $(25, 15, 9, 5, 3)$ when $k = 4$. Each $s_i \times s_i$ latent is partitioned into non-overlapping $p \times p$ patches, contributing $t_i = (s_i/p)^2$ memory tokens. Because $s_i$ decays geometrically, the total conditioning cost

$$T = \sum_{i=0}^{k} t_i = t_0 \left( 1 + \varphi^{-2} + \varphi^{-4} + \ldots \right) < \frac{\varphi^2}{\varphi^2 - 1} t_0 \approx 1.62 t_0$$

remains a constant factor above the single-frame cost $t_0$. Thus GoldenMem preserves a $(k+1)$-frame history with a fixed small overhead, as visualized in Figs. 3 and 4.

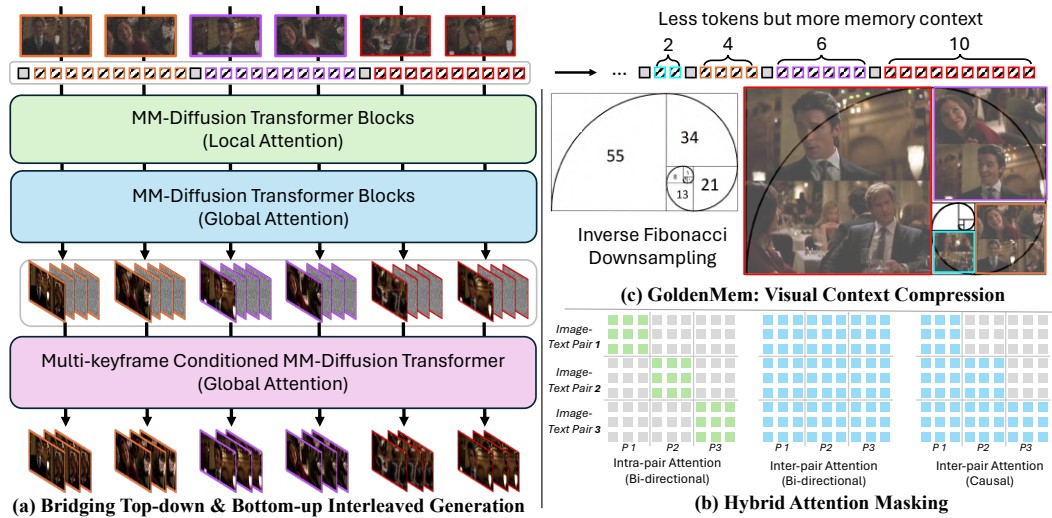

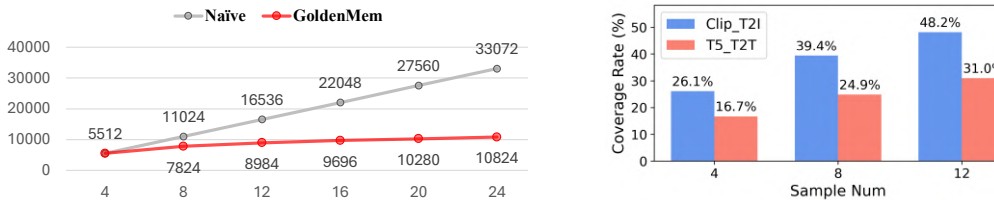

Figure 3: **Method Overview.** Captain Cinema bridges top-down and bottom-up interleaved for one-stage multi-scene movie generation. It introduces a hybrid attention masking strategy with GoldenMem context compression to learn and generate long movies efficiently and effectively. The number of GoldenMem tokens (referring to the short side of encoded image latents) is an example to show the inverse Fibonacci downsampling operation for long context compression.

Figure 4: **GoldenMem compresses the the context length.** The x-axis shows the number of image-text pairs, and the y-axis is the total number of tokens. Initial resolution is $400 \times 800$ (H×W).

Figure 5: **Semantic-oriented context retrieval.** CLIP text-to-image beats T5 text-to-text for history context retrieval for in-context storyline video extension.

**Semantic-Oriented Context Conditioning Beyond Temporal Distance.** Although sequential generation is intuitive, films frequently employ non-linear devices such as flashbacks, foreshadowing, and temporal loops. We therefore retrieve context by semantic relevance, conditioning each new frame on embeddings obtained with CLIP (text–image) and T5 (text–text). This strategy accommodates complex narrative structures more effectively than strict temporal conditioning. As shown in Fig. 5, CLIP Text-to-Image retrieval consistently yields higher coverage than T5-Text-to-Text retrieval, leading to better memory recall for extracting most relevant history frames.

**Progressively Finetuning with Growing Context Length.** Directly finetuning on long context interleaved sequences are prone to model collapse and often generate messy backgrounds with broken semantics (§4.2). To facilitate stable training over interleaved sequences, we use a progressive training strategy to finetune the model on interleaved data with growing context length. Specifically, we progressively finetune the model from single-image geneeration to 8, 16, and 32 interleaved pairs.

**Dynamic Stride Sampling for Interleaved Sequences.** Limited by the data scale, naively sample consecutive interleaved keyframes can lead models to get overfitted and be less robust. In this way, we use a dynamic stride sampling strategy to sample interleaved data, which provides thousands of times more valid data sequences (a bar of 25% overlap rate) than naive consecutive sampling.

### 3.3 BOTTOM-UP KEYFRAME-CONDITIONED VIDEO GENERATION

Our framework bridges two complementary viewpoints. From a *bottom-up* perspective, we begin with a base video generator and expand its scope to long-form content by conditioning on a sequence of interleaved shots. From a *top-down* perspective, we first construct a sparse set of narrative keyframes that act as visual anchors, then instruct the generator to fill the intervals between them. Unifying these perspectives allows us to preserve local visual fidelity while maintaining global nar-

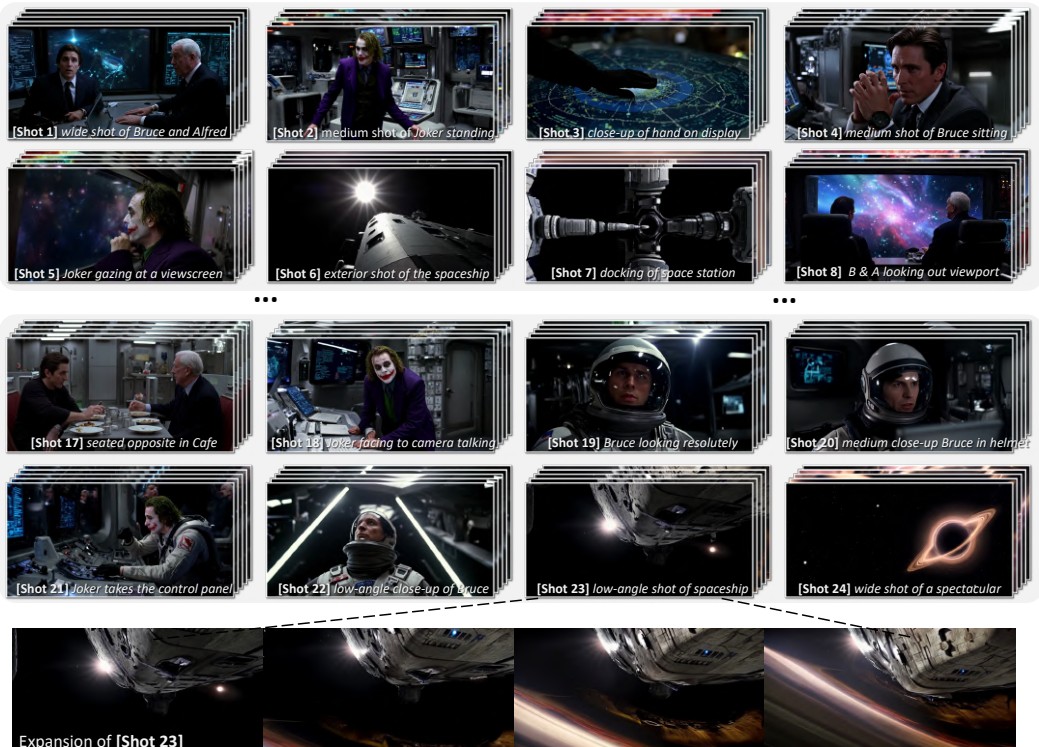

Figure 6: **Qualitative results.** From the narrative prompt "*An interstellar voyage with Bruce Wayne, the Joker, and Alfred Pennyworth,*" Gemini 2.5 Pro composes shot-level descriptions that guide our top-down key-frame generator, yielding the storyboard panels shown above. Each text–key-frame pair then conditions our bottom-up video model, which synthesises the full multi-scene film. The figure highlights twenty-four representative shots demonstrating sustained narrative coherence, character fidelity, and visual style across the entire production. Video demo is in the supplementary.

rative coherence throughout the entire video. Given $K$ keyframes $\{I_1, \ldots, I_K\}$, each the first frame of its shot, we condition a diffusion generator on (i) the tiled global caption $c_{\text{tiled}}$ and (ii) the visual embeddings of all keyframes:

$$V_k = \Big\{I_k, \ G_\theta\big(I_{1:K}, \ c_{\text{tiled}}, \ t = 2{:}T_k\big)\Big\}, \qquad k = 1, \ldots, K,$$

where $G_\theta$ outputs frames $f_{k,2}, \ldots, f_{k,T_k}$ for shot $k$. Multi-key conditioning thus anchors appearance at shot boundaries and enforces seamless motion dynamics across shots, generating videos that preserve narrative intent, visual details, and temporal coherence with strong visual consistency.

## 4 EXPERIMENT

**Experimental Setup.** We finetune Flux 1.Dev (Labs, 2024) on our interleaved movie dataset (described in §3.1) for interleaved keyframe generation. All keyframe models are trained for 40,000 steps with a batch size of 32 using 32 H100 GPUs. For interleaved video generation, we adopt Seaweed-3B (Seawead et al., 2025; Guo et al., 2025) as the base model and finetune it with multi-frame interleaved conditioning for 15,000 steps using 256 H100 GPUs. Additional implementation details are provided in the appendix. The keyframes are cropped with max area to 400x800 resolution without aspect ratio changes while the video generation data processing resizes all the videos to an approximation area of 230,040 total pixels. More implementation details of training our bottom-up and top-down keyframe and video generation models are provided in Section A.

### 4.1 MAIN RESULTS

**Qualitative results.** Fig. 6 shows a qualitative result with a storyline generated by Gemini 2.5. Specifically, we first prompt Gemini to generate a sequence of long-form captions describing a storyline inspired by Bruce Wayne, Alfred Pennyworth, and Joker, and interstellar travel. We then

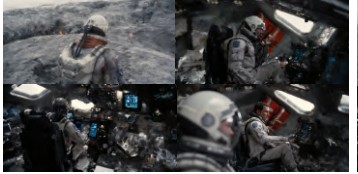 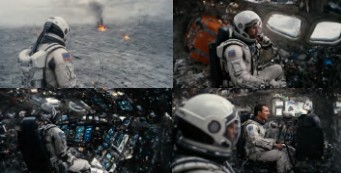 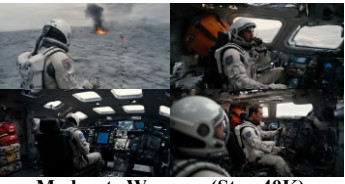

| **No Warmup** | **Over Warmup (Step 80K)** | **Moderate Warmup (Step 40K)** |

Figure 7: **Progressive long-context finetuning needs modest warmup.** As the base model (FLUX 1.Dev) is a distillation model with guidance, it requires progressive finetuning with growing context length to warmup the model. However, over warmup can also lead the model to forget the distilled knowledge for high-quality keyframe generation, generating artifacts or messy textures. Picking a proper warm-up length is therefore important for long context tuning.

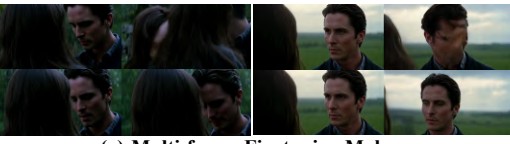 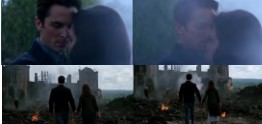 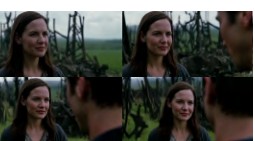

| **(a) Multi-frame Finetuning Makes Less Artifacts** | **(b) Context-aware "World" Generation** | **(c) Smooth Camera Movement** |

Figure 8: **Disentangled task modeling makes robust movie generation.** With the high-quality keyframes generated by our top-down keyframe generation model, the bottom-up video generation model can now focus on motion dynamics. The generation performs higher temporal consistency, world-context awareness and smoother camera movement.

use our interleaved top-down keyframe generation model to generate shot-wise keyframes based on the narrative. Finally, we construct interleaved text–keyframe pairs from the generated content and use them as conditioning input for the LCT generation model to produce the final movie. The results demonstrates compelling qualitative performance in the aspects of visual quality, consistency and semantic alignment with prompts. We also provide additional qualitative results of keyframe generation in Section G and also multi-scene movie results in the supplementary material.

| Method | Visual | | Temporal | | Semantic | User Study | |
|---|---|---|---|---|---|---|---|
| | Aesthetic↑ | Quality↑ | Consistency↑ | Dynamic↑ | Text↑ | Quality↑ | Semantic↑ |
| IC-LoRA (Huang et al., 2024a) | 54.1 | 60.5 | 88.7 | 61.1 | 23.1 | 1.5 | 2.3 |
| LCT (Guo et al., 2025) | 56.2 | 59.9 | **94.8**\* | 51.8 | 23.9 | 2.4 | 3.1 |
| **Ours (w/o MF-FT)** | 56.8 | 60.9 | 91.9 | 64.4 | 25.7 | 2.8 | 3.5 |
| **Ours** | **57.2** | **61.7** | 91.0 | **65.4** | **26.1** | **3.3** | **3.7** |

Table 1: **Quantitative Evaluation & User Study.** We employ automatic metrics and average human ranking (AHR). "Consistency" represents the average score of subject and background consistency. \*: Most video clips show low temporal dynamic but perform high TC score (Zheng et al., 2025).

**Quantitative Evaluation & User Study.** We follow VBench-2.0 (Zheng et al., 2025) to evaluate visual and temporal aspects, and follow the LCT (Guo et al., 2025) protocol for assessing text–semantic alignment. A user study is conducted using a 4-point scale—Very Good, Good, Poor, and Very Poor—focusing on semantic alignment and overall visual quality. As few works target long-video generation, we primarily compare against two closely related baselines: LCT and IC-LoRA (Huang et al., 2024a) combined with I2V (Seawead et al., 2025). To ensure a fair comparison, we use GPT-4o to format consistent scene prompts and generate videos across all methods. Tab. 1 shows that our approach performs favorably across most metrics, a finding corroborated by user studies evaluating video quality and semantic relevance. Notably, our advantages are most evident for generating coherent and vivid motion over long sequences in cinematic content creation.

**Long-context stress test.** We evaluate robustness as the context window grows from 8→48 interleaved pairs. For each setting we generate 20 clips and ask Gemini Flash 2.5 to score *Consistency*, *Quality*, *Diversity*, and *Narrative Coherence*; and another *Identity Consistency* metric used in VBench-2.0. As context length increases, LCT quality degrades sharply (Tab. 2), whereas ours retains $> 93\%$ of its consistency even at 48 pairs, validating the effectiveness of GoldenMem ( Fig. 11).

| Method | Ctx. Pairs | Consistency | | Visual Quality | | Diversity | Narrative | Identity |
|---|---|---|---|---|---|---|---|---|
| | | Character | Scene | Visual | Aesthetic | Diversity | Nar. Coher. | Identity |
| LCT (Guo et al., 2025) | 8 | 4.3 | 3.5 | 4.8 | 4.1 | 3.0 | 2.8 | 0.43 |
| LCT (Guo et al., 2025) | 16 | 3.6 | 3.4 | 4.5 | 3.9 | 2.5 | 2.8 | 0.31 |
| LCT (Guo et al., 2025) | 24 | 3.1 | 0.7 | 1.6 | 1.6 | 1.7 | 0.6 | 0.14 |
| **Ours** | 8 | 4.9 | 3.9 | 4.9 | 4.7 | 4.5 | 4.0 | 0.51 |
| **Ours** | 16 | 4.8 | 3.8 | 4.9 | 4.6 | 4.2 | 3.8 | 0.47 |
| **Ours** | 24 | 4.6 | 3.8 | 4.9 | 4.6 | 4.2 | 3.6 | 0.42 |
| **Ours** | 32 | 4.5 | 3.8 | 4.9 | 4.6 | 3.9 | 3.4 | 0.37 |
| **Ours (w/ G.Mem)** | 16 | 4.6 | 3.8 | 4.9 | 4.6 | 4.2 | 3.8 | 0.44 |
| **Ours (w/ G.Mem)** | 32 | 4.6 | 3.5 | 4.7 | 4.6 | 4.0 | 3.5 | 0.35 |
| **Ours (w/ G.Mem)** | 48 | 4.5 | 3.0 | 4.7 | 4.5 | 3.9 | 3.3 | 0.31 |

Table 2: **Long-Context Stress Test.** We exhibit long context stress test to benchmark the robustness of long context generation. We use Gemini 2.5 Flash to rate the generation quality in multiple aspects (detailed in Section H), and an automatic identity consistency metric introduced in VBench 2.0. Our interleaved method with GoldenMem can do high-quality long context generation with strong consistency preservation of both characters and scenes.

| # Pairs | GoldenMem | Compute | Visual | | Text |
|---|---|---|---|---|---|
| | # Groups × Images | PFLOPS (Est.) | Quality↑ | Consistency↑ | Alignment↑ |
| 16-frame | – | 30 | 4.4 | 4.7 | 4.1 |
| 16-frame | 3 × 4 | 21 | 4.3 | 4.3 | 4.1 |
| 32-frame | – | 55 | 4.1 | 4.3 | 3.8 |
| 32-frame | 3 × 8 | 35 | 3.9 | 4.2 | 3.6 |
| 48-frame | – | OOM | – | – | – |
| 48-frame | 3 × 12 | 52 | 3.6 | 4.2 | 3.5 |

Table 3: **GoldenMem makes longer context with less computation.** The first row of each block is the baseline while the second row is using GoldenMem. GoldenMem uses less computation with performance preservation and processes longer context window without out-of-memory (OOM). *Compute* is estimated through #tokens × steps × FLOP/token.

## 4.2 ABLATION STUDIES

**From Short to Long: Progressive Long-context Finetuning.** As detailed in §3.2, we employ progressive long-context fine-tuning, gradually expanding the context window of the interleaved keyframe generator. Fig. 7 shows that directly fine-tuning the FLUX base model leads to training collapse. On the other hand, fine-tuning from a late checkpoint (step 80,000; half of the target context length) also produces visual artifacts, which is likely caused by the knowledge forgetting for the FLUX distillation base model. So, progressive finetuning with FLUX-like distillation models needs a modest warmup (i.e., step 40,000) to avoid training collapse or forgetting distilled guidance.

**GoldenMem: Compressed Long-Context Memory.** As detailed in §3.2, we downsample historical visual context at the pixel level in inverse Fibonacci sequence, thereby capping the overall number of visual tokens. Tab. 3 ablates this design with different number of interleaved pairs. GoldenMem demonstrates strong compute efficiency under the same visual context, performing minor visual quality and consistency degradation. Moreover, GoldenMem enables longer context window (i.e., from 32 to 48 without OOM), which keeps longer context history for long video generation.

**Robust Long Video Generation with Disentangled Task Modeling.** As detailed in §3.2, we fine-tune a pretrained MM-DiT video generator (Seawead et al., 2025) with multi-frame interleaved conditioning to serve as our bottom-up component. Coupled with the top-down interleaved keyframe generator, this design improves robustness, yielding fewer artifacts (Fig. 8a), consistent environment dynamics—e.g., burning ruins with rising smoke (Fig. 8b)—and smooth camera motion with stable character identities (Fig. 8c). These results demonstrate that disentangling high-level narrative planning from low-level motion synthesis enables efficient and robust long-video generation.

**Dynamic Strided Data Sampling.** Because the scale of our movie dataset is relative small to the data used for base-model pre-training, naïve sampling induces rapid overfitting and weak generalization, particularly under sparse keyframe conditioning and limited interleaved samples. We

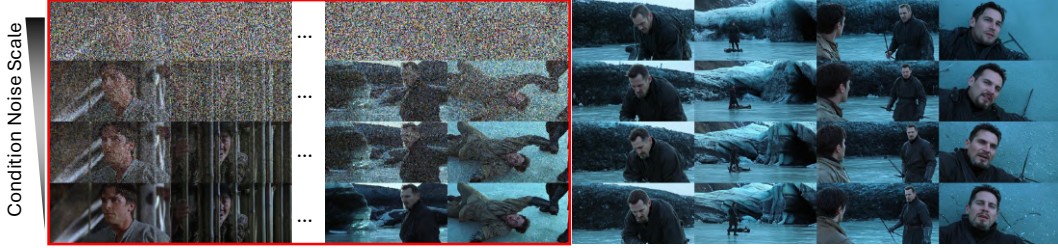

Figure 9: **Visual context conditioning needs moderate noise injection.** Each conditioning frame has individually been injected with random noise at varying levels: *1st*: 751∼1000; *2nd*: 501∼750; *3rd*: 251∼500; *4th*: 1∼250. Context conditioning with moderate noise lead to better consistency.

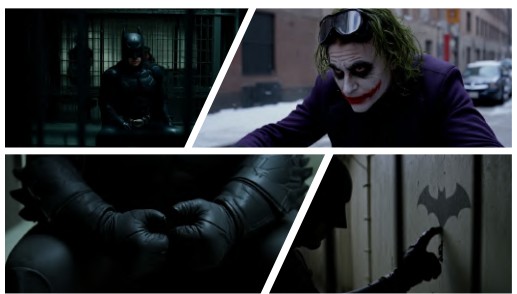

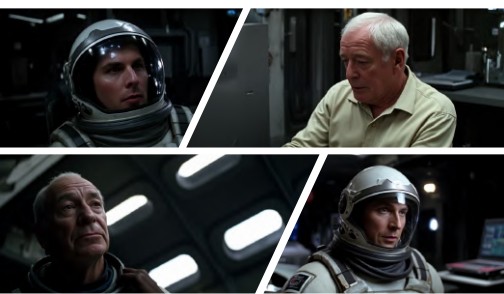

(a) **Creative scene generation.** *Batman is locked in prison while Joker takes control of Gotham.*

(b) **Cross-movie character swapping.** *Bruce Wayne and Alfred Pennyworth appear in "Interstellar."*

Figure 10: **Captain Cinema generalization abilities.** Trained on high-quality, richly annotated interleaved data, our model offers creative control at the movie, scene, and character levels, allowing limitless "parallel universes" that maintain coherent narratives and high visual fidelity.

mitigate this by introducing a dynamic strided sampling scheme that systematically offsets the sampling stride across epochs. This strategy yields 100 times more valid data when sample 16-frame sequence from 32 frames (25% keyframe-overlap threshold).

**Noisy visual context conditioning.** During our interleaved keyframes generation model training, each frame is applied with individual level of noise. So, we ablate the noise level for keyframe extending generation with a given context of noisy latents. As shown in Fig. 9, our interleaved keyframe generation model fails to keep character consistency when conditioning on context with large noise injection (i.e. 501∼1000 steps). However, our model shows very strong character consistency when using moderate noise level (i.e. 1∼500 steps) and shows best character facial perservation when using a small noise (i.e. adding 1∼250 timesteps of noise) for long context conditioning. We also evaluate out-of-domain generalization to unseen characters in Section I.

### 4.3 GENERALIZATION ABILITIES

**Creative Scene Generation.** Our method could synthesize novel scenes that were never present in the training corpus by recombining familiar characters, settings, and plot elements. Leveraging semantic-oriented retrieval and interleaved conditioning, it produces imaginative environments, e.g., a prison-bound Batman confronting the Gotham under Joker's control, while preserving visual fidelity and narrative plausibility through one-stage generation. Our model effectively understands character traits, typical settings, and plausible interactions, enabling seamless integration of new creative ideas into cohesive, realistic scenes. Our model could extend any scene through context conditioning and continue story-telling with characters and scenes consistency.

**Cross-Movie Character Swapping.** Our pipeline also enables seamless swapping of characters across unrelated cinematic universes. Leveraging identity-preserving embeddings, the system can insert Bruce Wayne and Alfred Pennyworth into the science-fiction environment of *Interstellar*. This capability demonstrates a clear separation between character identity and scene context, providing a flexible platform for counterfactual narrative exploration.

## 5 CONCLUSION

We propose Captain Cinema, based on top-down interleaved keyframe planning with bottom-up, multi-keyframe-conditioned video synthesis to generate a short movie. Leveraging the Golden-Mem compressed visual context, progressive long-context fine-tuning, and dynamic strided sampling training strategies, our model sustains global narrative coherence while preserving local visual fidelity throughout feature-length videos. Captain Cinema also demonstrates generalization ability for creative scene generation and cross-movie character swapping. Despite the limitations discussed above, our work represents a concrete step toward fully automated, story-driven movie generation.

## ETHICS STATEMENT

Long-form video generation could democratize high-quality animation, documentary creation, education, and simulation, enabling content production that previously demanded extensive budgets and expertise. This technology also provides valuable tools for individuals with limited mobility or resources and offers new avenues for reinforcement learning and robotics simulations. However, accessible video generation also raises concerns around hyper-realistic misinformation, non-consensual media, intellectual-property violations, and environmental impact due to high computational costs. To mitigate these risks, we will adopt gated model releases, implement robust watermarking, provide model documentation, and enforce clear usage policies. Additionally, we will develop watermark verification tools, conduct external red-team audits, and collaborate with stakeholders on content detection benchmarks. All the data here are used for research only purpose.

## REPRODUCIBILITY STATEMENT

We include all the key details including training hyperparameters, data collection and processing procedure, evaluation and annotation prompt etc.

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

APPENDIX

Please visit the webpage in supplementary material for viewing a set of video demos and understand the model capabilities.

## A  IMPLEMENTATION DETAILS

We provide the training hyperparameters for the top-down keyframe generation model and the interleaved conditioned video generation model in Tab. 4. The interleaved auto-regressive model is trained on interleaved image-text pairs with a default resolution of $400 \times 800$, using a batch size of 32 and bfloat16 precision. It employs AdamW as the optimizer, with a peak learning rate of $2 \times 10^{-4}$ and a cosine decay schedule, training for 2,500 steps. Training context pairs vary between 2 and 8, while inference always uses 8 pairs for consistency. The visual-conditioned video generation model processes video data at an area of 480x480 ( 230,400 total pixels), with a batch size of 64 and bfloat16 precision. It uses AdamW with a peak learning rate of $1 \times 10^{-5}$ and a constant decay schedule, training for 20,000 steps to handle temporal conditioning effectively. We provide the training hyperparameters for the top-down keyframe generation model and the interleaved conditioned video generation model in Tab. 4 The interleaved auto-regressive model is trained on interleaved image-text pairs with a default resolution of $400 \times 800$, using a batch size of 32 and bfloat16 precision. It employs AdamW as the optimizer, with a peak learning rate of $2 \times 10^{-4}$ and a cosine decay schedule, training for 2,500 steps. Training context pairs vary between 2 and 8, while inference always uses 8 pairs for consistency. The visual-conditioned video generation model processes video data at an area of 480x480 ( 230,400 total pixels), with a batch size of 64 and bfloat16 precision. It uses AdamW with a peak learning rate of $1 \times 10^{-5}$ and a constant decay schedule, training for 20,000 steps to handle temporal conditioning effectively.

| Configuration | Keyframe Generation Model | Video Generation Model |
|---|---|---|
| Resolution | $400 \times 800 \times L$ | $(480P, \text{ native AR}) \times T$ |
| Optimizer | AdamW | AdamW |
| Optimizer hyperparameters | $\beta_1{=}0.9,\ \beta_2{=}0.999,\ \epsilon{=}10^{-8}$ | $\beta_1{=}0.9,\ \beta_2{=}0.95,\ \epsilon{=}10^{-8}$ |
| Peak learning rate | $4 \times 10^{-4}$ | $1 \times 10^{-4}$ |
| Learning-rate schedule | Linear warm-up, cosine decay | Linear warm-up, cosine decay |
| Gradient clip | 1.0 | 1.0 |
| Training steps | 20,000 (per stage) | 10,000 (total) |
| Warm-up steps | 500 | 1,000 |
| Batch size | 16 | $\approx 64$ |
| Numerical precision | bfloat16 | bfloat16 |
| Computation | $32 \times$ H100, $\sim$72 h | $256 \times$ H100, $\sim$200 h |

Table 4: **Implementation details of our models.** The left column lists the interleaved keyframe generator settings, and the right column lists the visual-conditioned video generator settings.

## B    DATA ANNOTATION PROMPT

You are a *film-director assistant* annotating a single frame from the movie.

1. **Detailed Caption**
   - Describe visual composition, lighting, and camera angles.
   - Note character actions, expressions, and positioning.
   - Mention setting details and atmosphere.

2. **Character Identification**
   - Identify characters *only when absolutely certain*, enclosing names in angle brackets (e.g. ¡Character Name ). Allowed list: *¡Character List¿*.
   - If uncertain, describe by appearance, role, or action—*do not* use angle brackets.
   - For dual-identity roles, choose the name that matches the on-screen persona.

3. **Level of Detail**
   - Craft concise yet thorough sentences capturing technical aspects (framing, lighting) and narrative elements (interactions, mood).
   - Focus on specifics that make the frame distinctive.

## C    THEORETICAL ANALYSIS OF GOLDENMEM

GoldenMem can be interpreted as a budgeted multi-scale memory over the history of keyframes. Let the newest frame be encoded at short side $s_0$ and let the $i$-th previous frame be stored at scale $s_i = \lfloor s_0/\phi^i \rfloor$, where $\phi = (1 + \sqrt{5})/2$. Up to integer rounding and the minimum latent size, the corresponding token count satisfies $t_i \approx t_0 \phi^{-2i}$ because the latent is partitioned into fixed-size patches. A naive memory that stores $n$ frames at full resolution requires $T_{\text{naive}}(n) = nt_0$ tokens, while GoldenMem requires

$$T_{\text{GM}}(n) = \sum_{i=0}^{n-1} t_i \approx t_0 \sum_{i=0}^{n-1} \phi^{-2i} = t_0 \frac{1 - \phi^{-2n}}{1 - \phi^{-2}} < \phi t_0 \approx 1.618 t_0.$$

Hence, GoldenMem turns linear memory growth into a constant-factor overhead over the most recent full-resolution frame. Since the dominant attention cost is quadratic in the number of conditioning tokens, the memory-dependent attention complexity drops from $O((nt_0)^2)$ to $O((\phi t_0)^2)$. The qualitative implication is that GoldenMem allocates fidelity according to recency: recent frames preserve dense spatial detail, whereas older frames retain coarse but semantically useful scene and identity anchors. Tab. 3 is consistent with this analysis, showing that the same design extends the context window from 32 to 48 frames without out-of-memory while maintaining similar generation quality.

## D    FURTHER EXPLANATION ON CONTEXT SELECTION AND SEMANTIC RETRIEVAL

For each target keyframe with caption $y_t$, we build a candidate set from all previous memory entries $\{(x_j, y_j)\}_{j<t}$ and rank them by semantic relevance instead of pure temporal distance. The default score is CLIP text-to-image similarity,

$$r_j = \text{sim}(f_{\text{CLIP}}^{\text{text}}(y_t), f_{\text{CLIP}}^{\text{img}}(x_j)),$$

and the selected history is the highest-scoring subset that fits the GoldenMem budget. To avoid saturating the memory with redundant frames, we suppress obvious near-duplicates from the same shot and retain context that covers complementary visual states. This design addresses a common ambiguity in long movies: captions repeatedly mention the same characters, camera types, or emotions, so several earlier frames may be lexically compatible with the current caption even when they belong to different scenes. Pure text-to-text retrieval can therefore over-match repeated wording and return a semantically related but visually mismatched scene. By grounding the query against

historical images, CLIP retrieval better disambiguates recurring characters and scene revisits while still allowing non-local context such as flashbacks or foreshadowing. When several candidates are nearly tied, we only use recency as a tie-breaker, which preserves local continuity without collapsing the retrieval to a purely temporal heuristic.

# E    LONG CONTEXT STRESS TEST

This is to provide the additional charts for the long context stress test. We evaluate robustness as the context window grows from 8→48 interleaved pairs. For each setting we generate 20 clips and ask Gemini Flash 2.5 to score *Consistency*, *Quality*, *Diversity*, and *Narrative Coherence*; and another *Identity Consistency* metric used in VBench-2.0. As context length increases, LCT quality degrades sharply (Tab. 2), whereas ours retains $> 93\%$ of its consistency even at 48 pairs, validating the effectiveness of GoldenMem (Fig. 11).

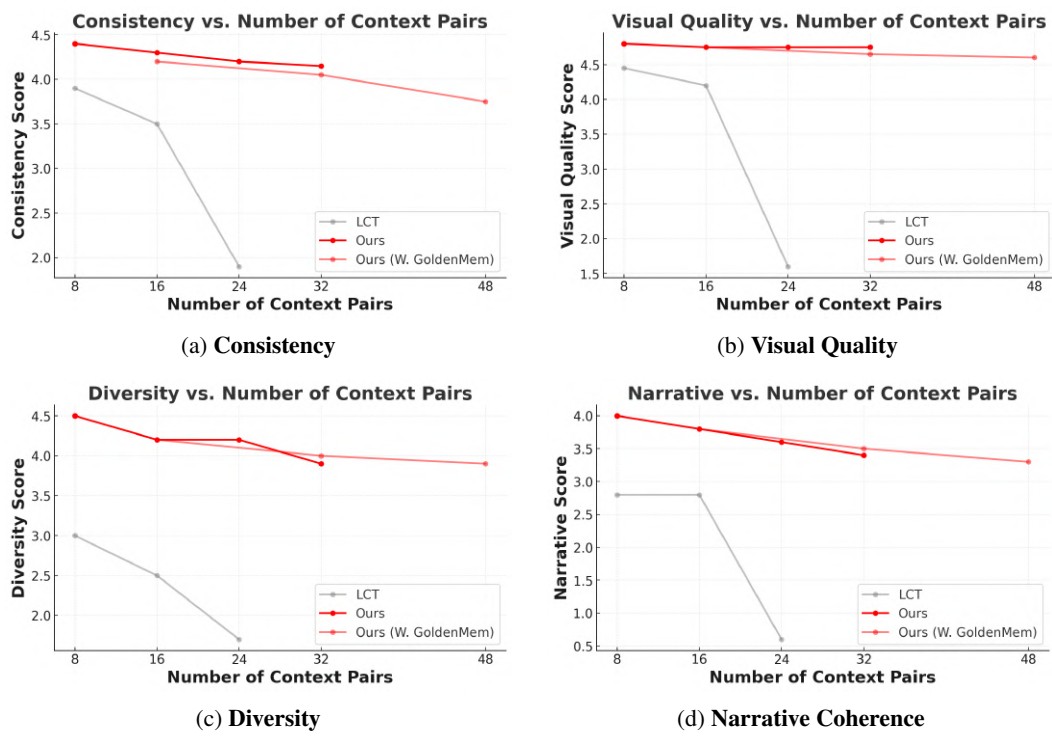

(a) **Consistency**                    (b) **Visual Quality**

(c) **Diversity**                    (d) **Narrative Coherence**

Figure 11: **Long-Context Stress Test.** While context length grows to be very long, our method still maintains strong visual consistency, high visual quality, diversity, and narrative coherence even under extended context lengths.

# F    HYBRID ATTENTION MASKING IN TRAINING AND INFERENCE.

The local mask is fixed across training and inference: the first $D$ double-stream blocks always use a block-diagonal mask, so each image-text pair performs within-pair fusion before any cross-pair interaction. The distinction appears only in the subsequent $S$ single-stream blocks. During training, we use a fully bidirectional global mask because the entire interleaved sequence is available and the objective is to learn symmetric long-range dependencies across all pairs. During inference, the model supports two regimes. For offline storyboard generation, where the captions of the whole sequence are given in advance and all frames are sampled jointly, we keep the same bidirectional global mask and perform diffusion-forcing style denoising; this is the setting used for the 32-frame qualitative examples in Appendix Sec. G. For online extension or autoregressive movie continuation, we replace the global mask with an upper-triangular causal mask so that the current pair only attends to previously generated pairs and never to future ones. No causal masking is applied inside the local double-stream blocks because causality is relevant only to cross-pair dependencies, not to fusion within a fully observed pair. Thus, the architecture is unchanged between training and inference; only the global inter-pair mask is switched according to whether future pairs are available.

## G ADDITIONAL QUALITATIVE RESULTS FOR KEYFRAME GENERATION.

We provide **non-cherry pick** qualitative keyframe generation results in figures below.

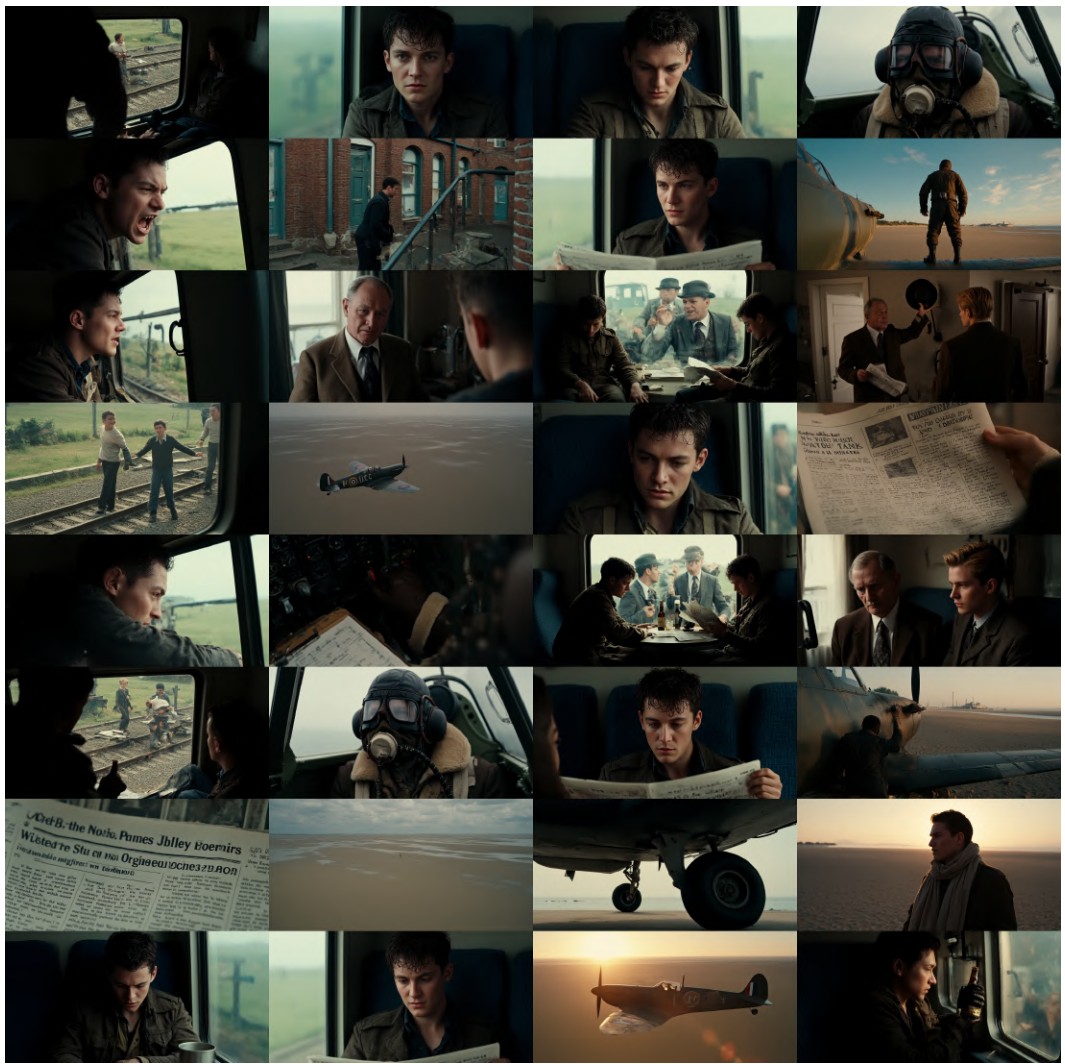

Figure 12: **Qualitative Result.** Our method shows superior generation quality with character and scene consistency. 32 frames are generated by our method through diffusion forcing with bidirectional masking strategy. The prompt here is from the validation split with ChatGPT-4o re-imagined.

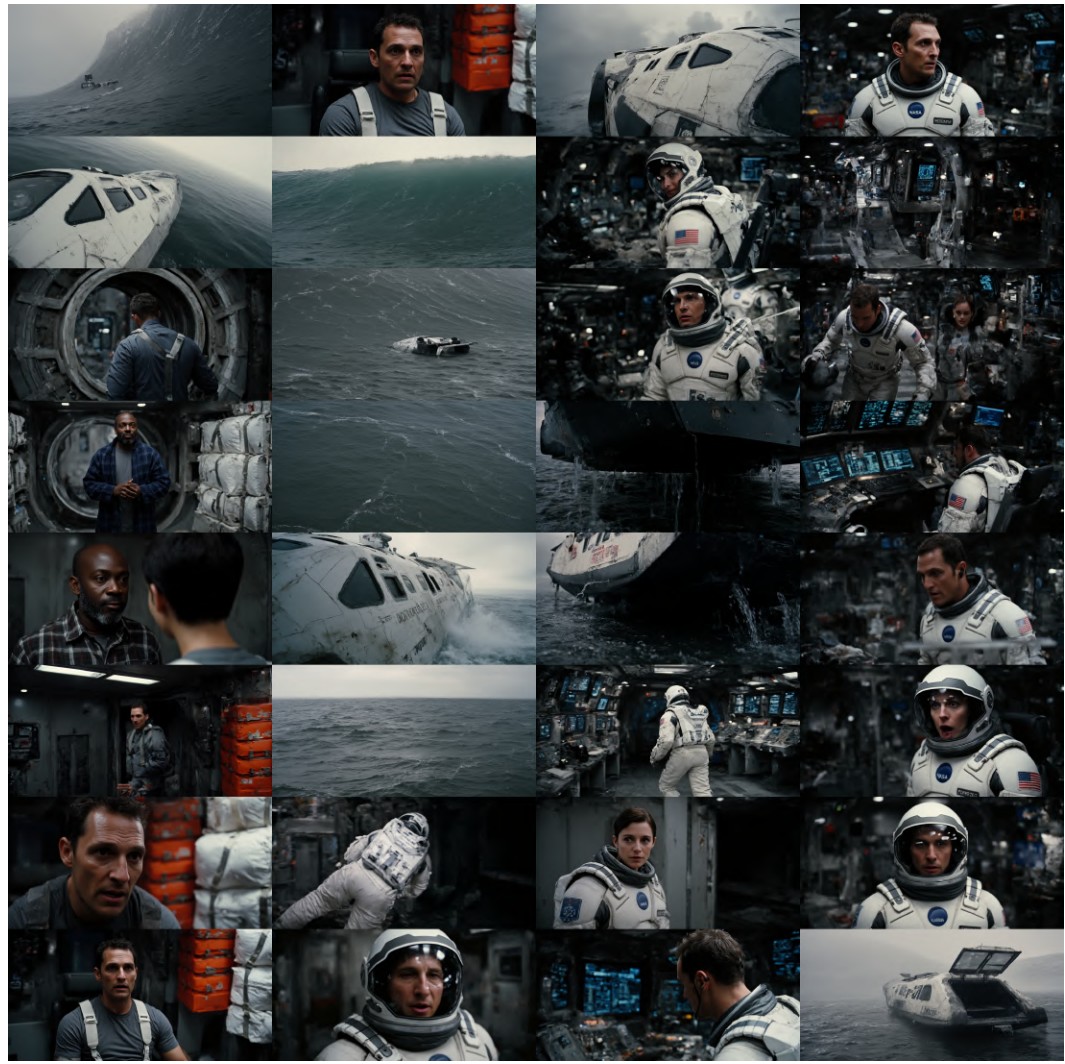

Figure 13: **Qualitative Result.** Our method shows superior generation quality with character and scene consistency. 32 frames are generated by our method through diffusion forcing with bidirectional masking strategy. The prompt here is from the validation split with ChatGPT-4o re-imagined.

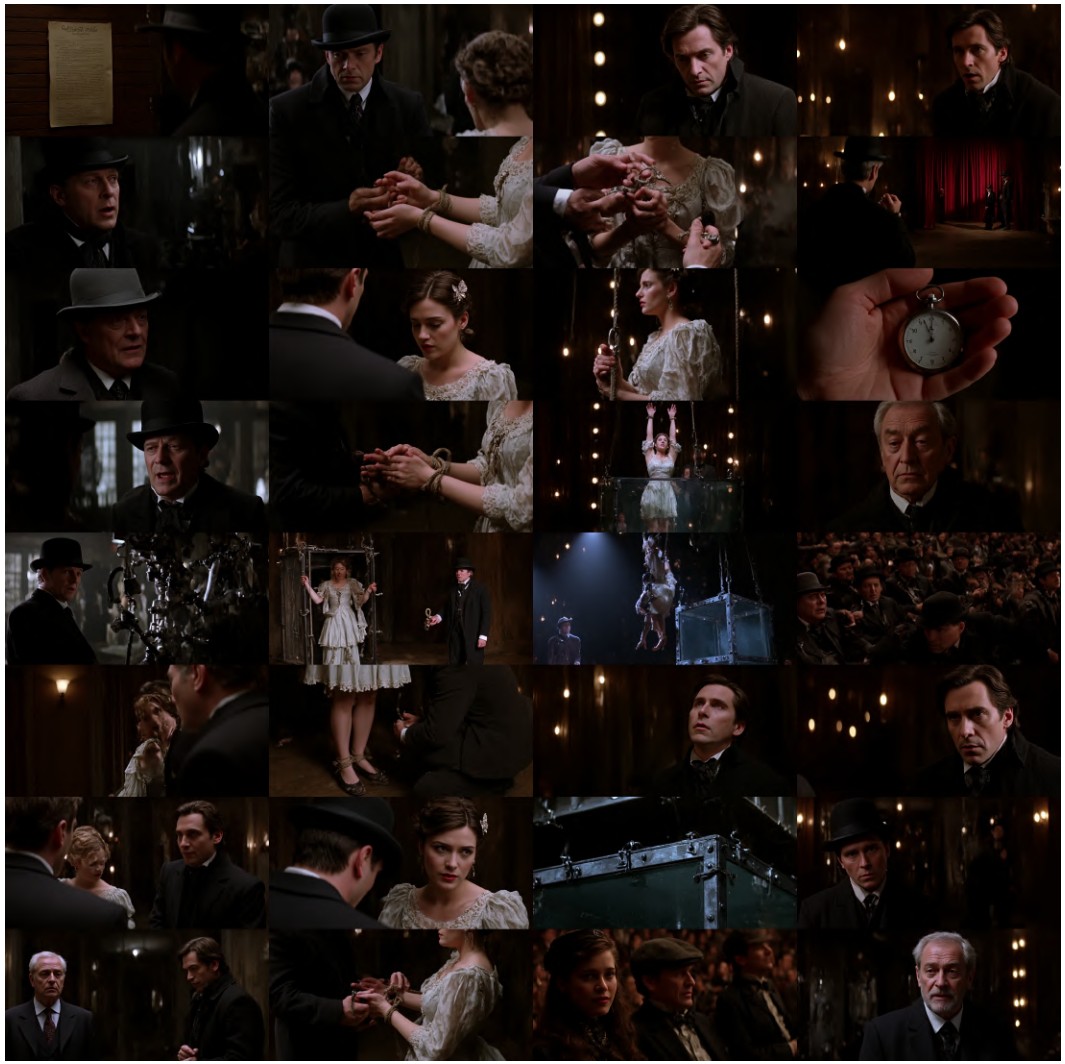

Figure 14: **Qualitative Result.** Our method shows superior generation quality with character and scene consistency. 32 frames are generated by our method through diffusion forcing with bidirectional masking strategy. The prompt here is from the validation split with ChatGPT-4o re-imagined.

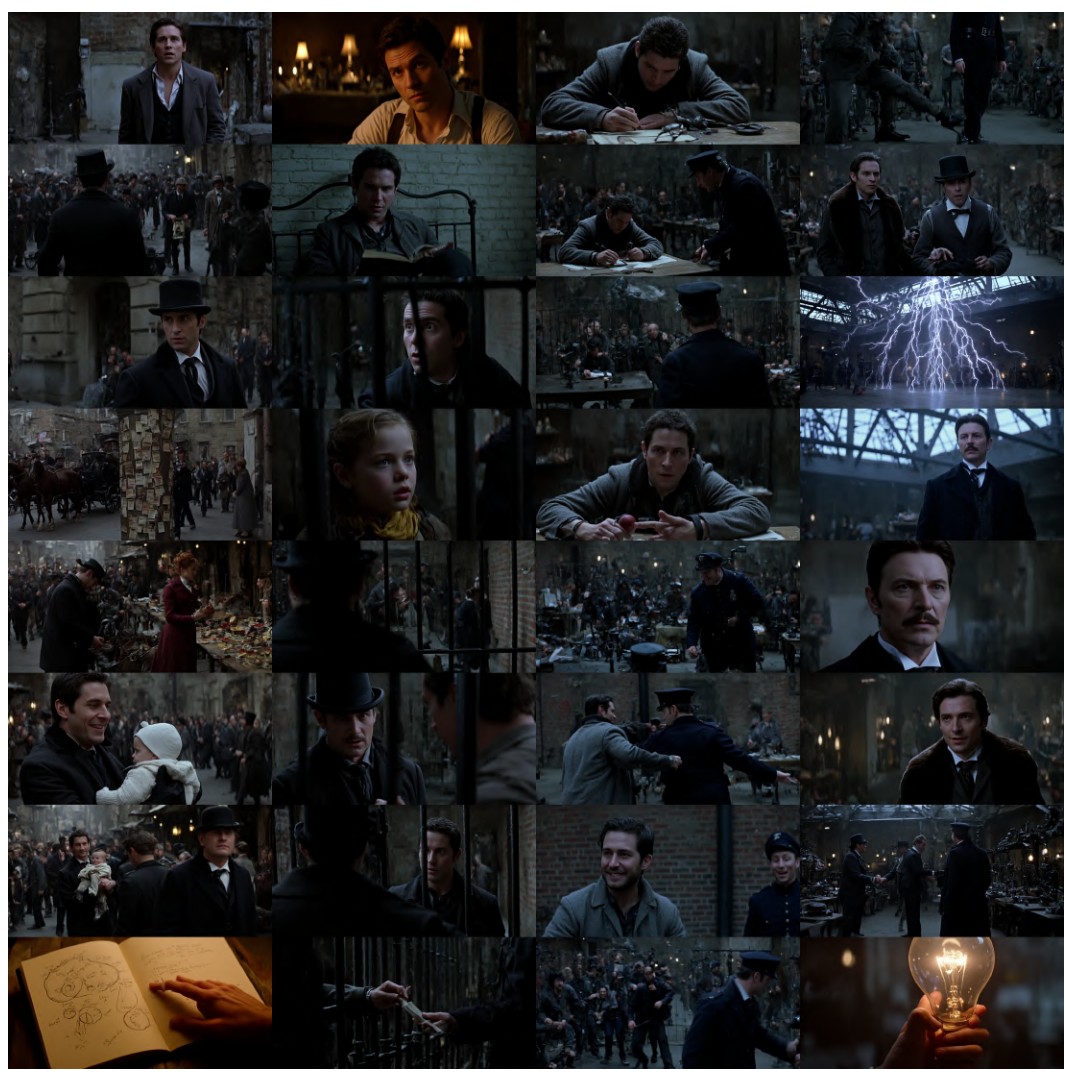

Figure 15: **Qualitative Result.** Our method shows superior generation quality with character and scene consistency. 32 frames are generated by our method through diffusion forcing with bidirectional masking strategy. The prompt here is from the validation split with ChatGPT-4o re-imagined.

## H    LONG-CONTEXT STRESS TEST DETAILS

We evaluate the robustness of our model under long-range dependencies by scoring sequences of keyframes—both ground-truth and those sampled from our generated clips— with Gemini 2.5 Flash. Guided by the system prompt in Section H, the evaluator rates each sequence along six dimensions: character consistency, scene consistency, visual quality, aesthetics, diversity, and narrative coherence. An overview of the per-aspect trends is visualised in Fig. 11, and the complete results for a range of context lengths are reported in Tab. 2.

```
SYSTEM_PROMPT

You are an expert visual analyst.
You will receive a sequence of movie keyframes (static images)
and must evaluate them in six aspects:

1. Character Consistency
2. Scene Consistency
3. Visual Quality
4. Aesthetics
5. Diversity (no duplicate or near-duplicate frames)
6. Narrative Coherence (does the sequence form a sensible mini-story?)

For each aspect, assign a 0{5 score and provide a one-sentence

justification.

Score meanings:
| Score | Meaning                                      |
|-------|----------------------------------------------|
| 0     | Unacceptable: completely fails expectations |
| 1     | Very Poor: major problems                    |
| 2     | Poor: noticeable flaws                       |
| 3     | Fair: meets basic expectations               |
| 4     | Good: solid, minor imperfections             |
| 5     | Excellent: flawless or near-perfect          |

Scoring rules
- Character Consistency: clothing, hairstyle, facial features,

and proportions
remain constant.
- Scene Consistency: lighting and background elements stay coherent

across
frames.
- Visual Quality: frames are sharp, well-exposed, and color-balanced.
- Aesthetics: composition, color palette, and contrast are deliberate
and pleasing.
- Diversity: frames are meaningfully distinct; penalize repetition.
- Narrative Coherence: frames suggest a logical progression; penalize
abrupt shifts.

Return valid JSON in the following structure:

{
  "character_consistency": { "score": <0-5>, "justification": "<text>" },
  "scene_consistency":     { "score": <0-5>, "justification": "<text>" },
  "visual_quality":        { "score": <0-5>, "justification": "<text>" },
  "aesthetics":            { "score": <0-5>, "justification": "<text>" },
  "diversity":             { "score": <0-5>, "justification": "<text>" },
  "narrative_coherence":   { "score": <0-5>, "justification": "<text>" }
}
```

## I  Out-of-domain Identity Preservation Generalizability

We assess our model's ability to preserve unseen identities under noisy conditioning. For each out-of-domain individual, we collect 16 photographs with accompanying captions. Twelve images serve as interleaved context; their visual embeddings are perturbed with Gaussian noise for robust conditioning. The remaining four captions are then used to prompt the model, which must synthesize four new frames of the same person. Qualitative results are shown in Figs. 16 and 18 (generation with different dressing) and Figs. 17 and 19 (generation with different dressing). Our interleaved keyframe generator effectively maintains identity and accurately follows the given prompting instruction (appearance descriptions), demonstrating strong generalization to unseen characters.

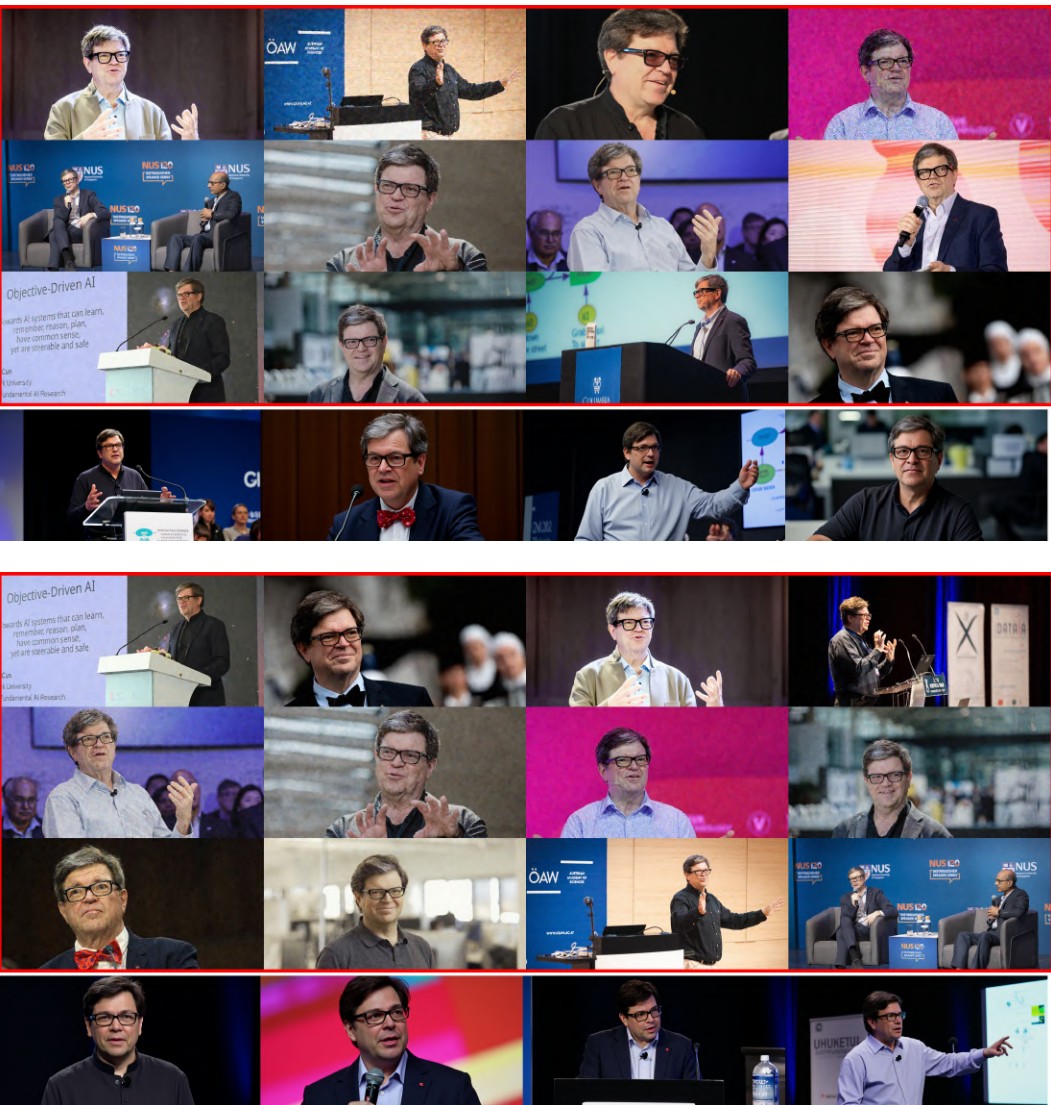

Figure 16: **Out-of-domain Identity-Preservation Test : Different Dressing.** Each test case begins with 12 randomly sampled context images that are injected with noise; the model then generates another four frames conditioned on textual prompts as well as the pretext interleaved conditions. Our interleaved key-frame generator accurately maintains the character's identity, demonstrating strong out-of-domain generalizability.

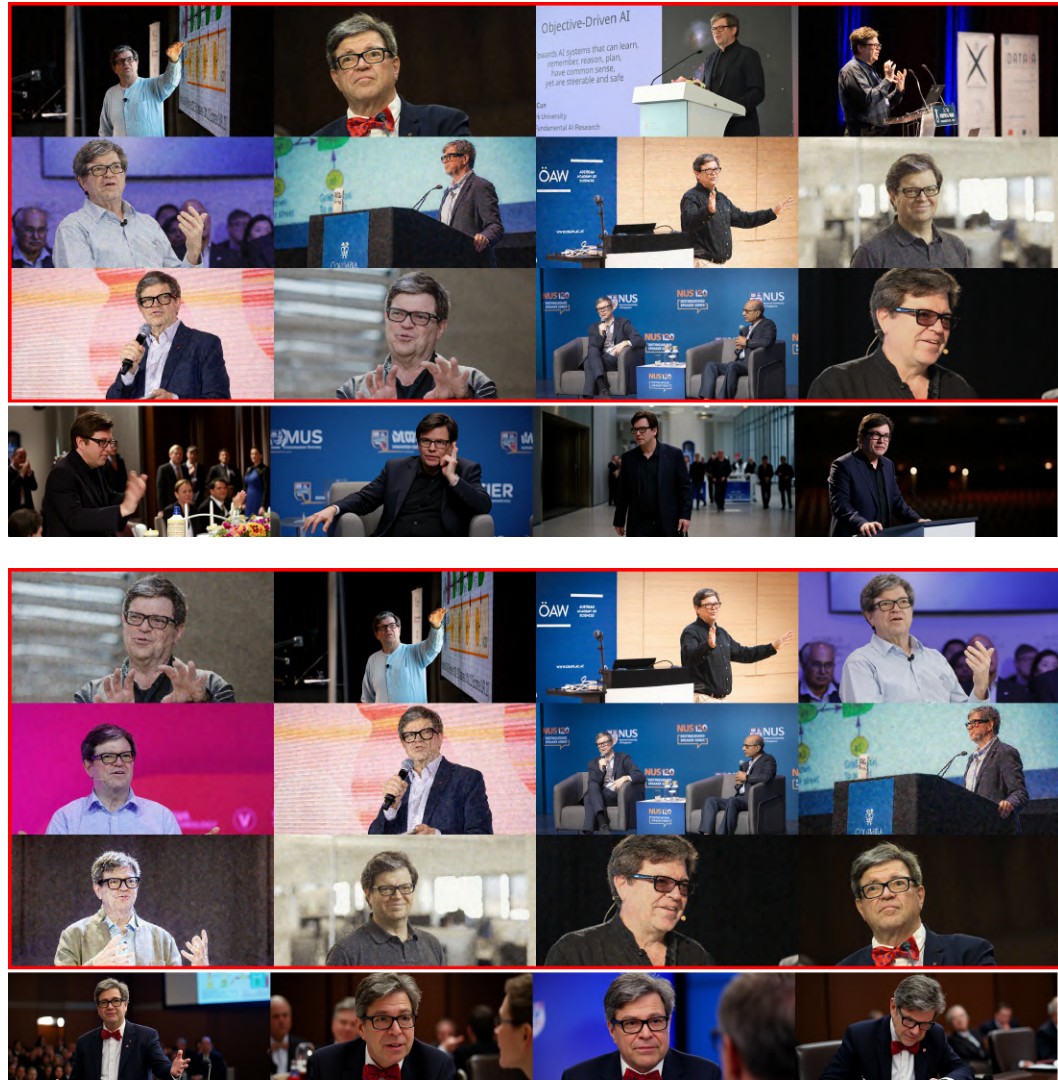

Figure 17: **Out-of-domain Identity-Preservation Test : Same Dressing.** Each test case begins with 12 randomly sampled context images that are injected with noise; the model then generates another four frames conditioned on textual prompts as well as the pretext interleaved conditions. Our interleaved key-frame generator accurately maintains the character's identity, demonstrating strong out-of-domain generalizability.

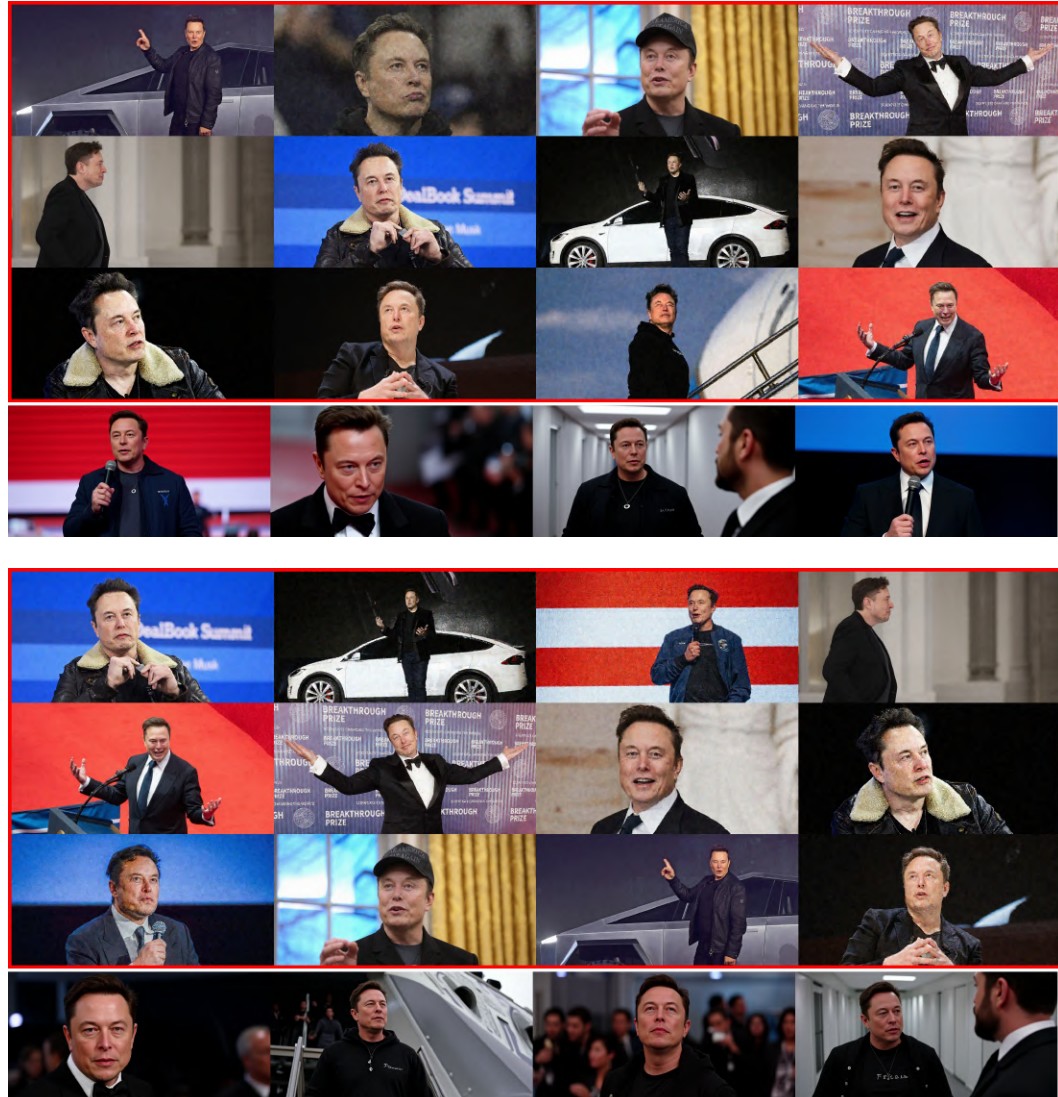

Figure 18: **Out-of-domain Identity-Preservation Test : Different Dressing.** Each test case begins with 12 randomly sampled context images that are injected with noise; the model then generates another four frames conditioned on textual prompts as well as the pretext interleaved conditions. Our interleaved key-frame generator accurately maintains the character's identity while follows the instruction prompts, demonstrating strong identity preservation ability to out-of-domain characters.

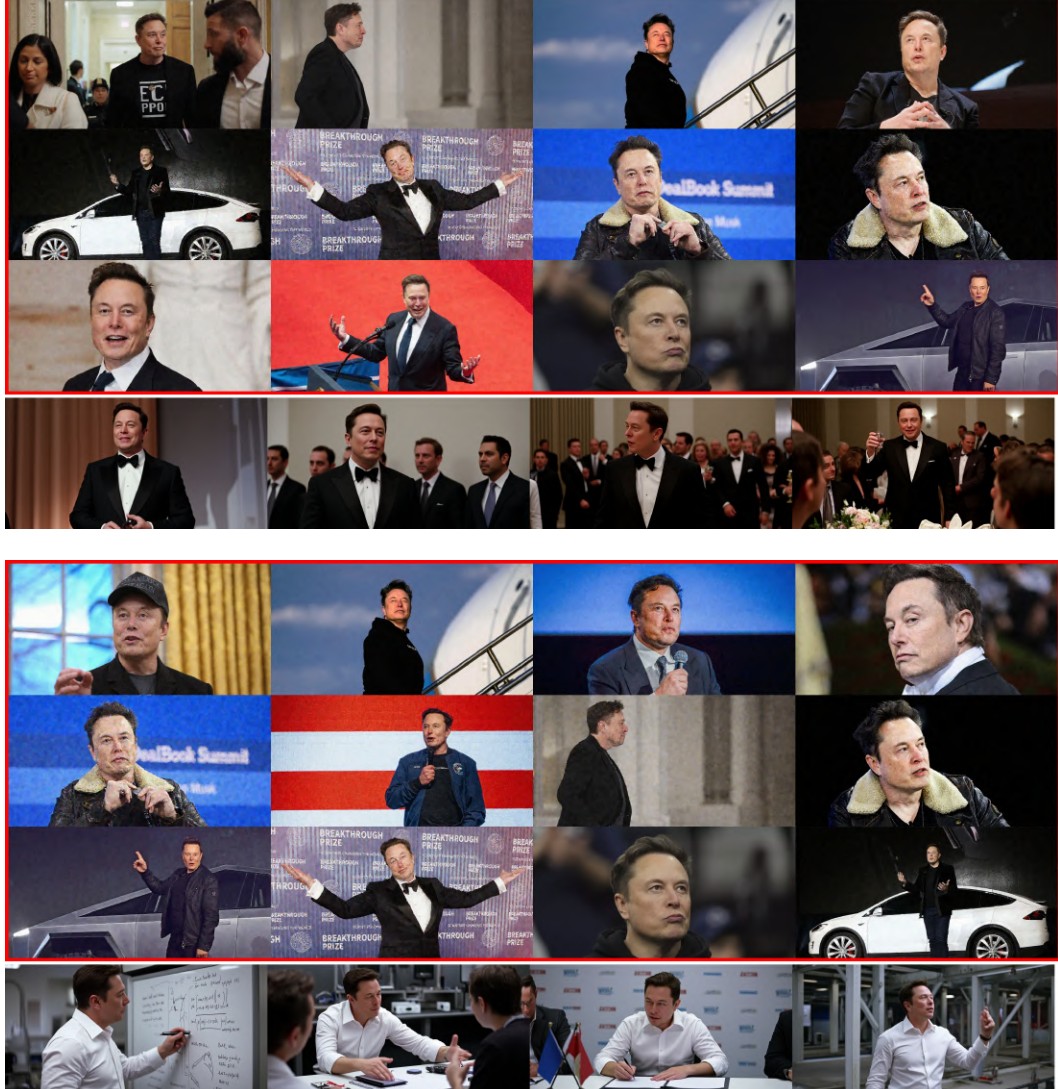

Figure 19: **Out-of-domain Identity-Preservation Test : Same Dressing.** Each test case begins with 12 randomly sampled context images that are injected with noise; the model then generates another four frames conditioned on textual prompts as well as the pretext interleaved conditions. Our interleaved key-frame generator accurately maintains the character's identity, demonstrating strong out-of-domain generalizability.

## J  DIFFERENCE TO FRAMEPACK

Framepack (Zhang & Agrawala, 2025) is a concurrent work on arXiv (confirmed with the authors), which proposes a similar token compression mechanism through context history down-sampling with exponential ratios (e.g., 1, 2, 4 ...). Below are explanations to several major difference in the aspects of motivation, applied scenario, technical design and empirical performance.

**Motivation.** While we share a similar goal to reduce the GPU memory usage, Framepack motivation is to enable consumer level GPU to be capable of large video generation model finetuning. Our movitation is to extend longer context as memory for preserving visual consistency.

**Applied scenario.** Framepack still focuses on classic single video clip generation and the training task is like image editing at first (generate last frames) and then do first-last frame video interpolation, which is illustrated in Fig. 20. Framepack supports motion difference with a major entity across frames but do not support scene moving or even scene changing. In contrast, our method is designed for multi-scene long video generation and the GoldenMem focuses on discrete keyframe generation whose task is more complex and challenging frame-level generation, illustrated in Fig. 21.

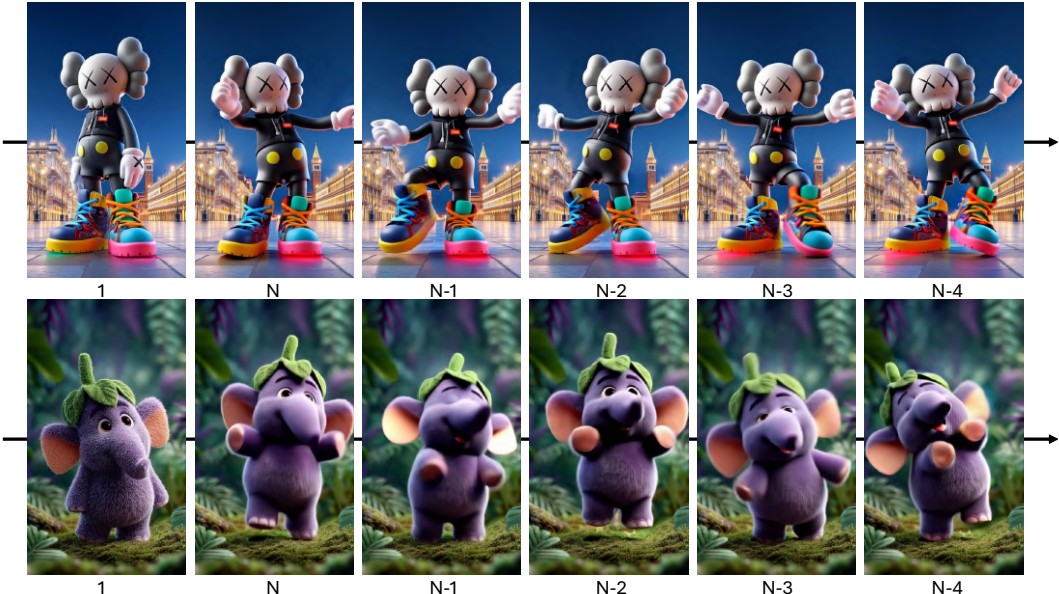

Figure 20: **Framepack firstly edits, then interpolates.** Framepack still focuses on classic single-clip video generation but in an anti-drifted auto-regressive way – firstly generates last frames, and then interpolates the middle frames given the context of the first frame and last frames.

**Technical design.** Framepack uses aggressive compression rate for context compression (downsample ratio is 2 or 4) with larger group sizes and introduces additional convolution kernels for the compression operation. In contrast, we use a fixed number frames for each group of context and use the fibonacci ratio ($\sim 1.618$) for history downsampling in pixel space directly.

**Empirical performance.** We follow the typical context compression setup proposed in Framepack `f1k1f2k2f16k4` for an empirical comparison to replace our GoldenMem compression setup under the 32-frame generation setting. As Framepack's downsampling ratio and strategy is more aggressive, we see worse visual consistency and visual quality than our GoldenMem setup and even slower convergence. So, we believe GoldenMem design is currently a more optimal solution for long-context multi-scene keyframe generation in a discrete way.

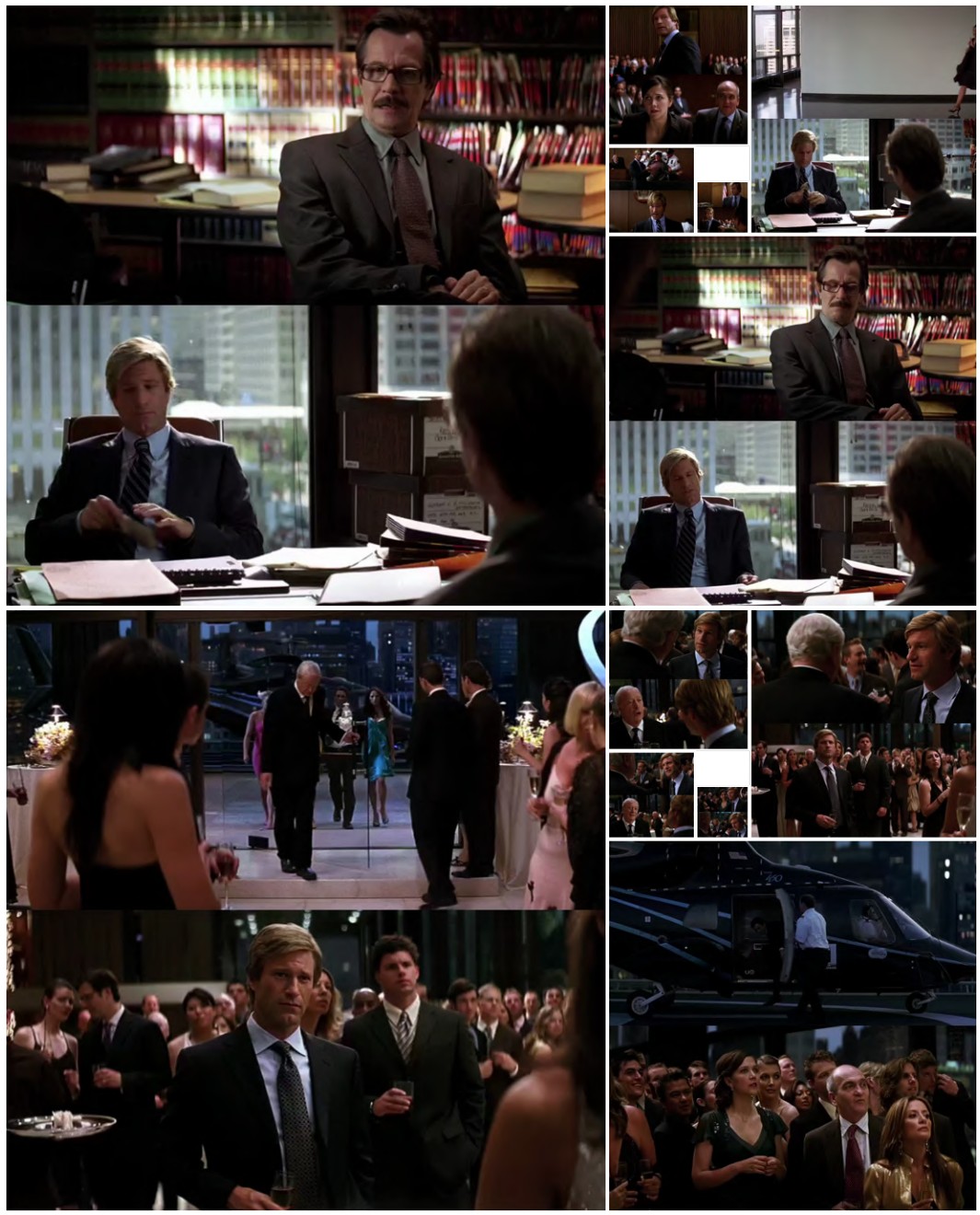

Figure 21: **GoldenMem is designed for multi-scene discrete generation.** GoldenMem needs to understand and align different scenes and characters for keeping visual consistency and also keep high visual quality. In contrary to continuous frame interpolation, our method is for discrete cross-scene keyframe generation which is more complex and challenging.

