# OpenReview forum: "Captain Cinema: Towards Short Movie Generation"
_ICLR.cc/2026/Conference — ICLR 2026 Poster_

### Official Review · Reviewer_TXMh · 2025-10-28

**Soundness:** 2
**Presentation:** 3
**Contribution:** 2
**Rating:** 4
**Confidence:** 4

**Summary:**

This paper presents Captain Cinema, a framework for generating short, narrative-driven movies from a detailed textual description. The method employs a two-stage process: a "top-down" planning step that generates a sequence of keyframes to ensure narrative and visual coherence, followed by a "bottom-up" video synthesis step that generates the motion between these keyframes. The authors introduce a visual context compression technique called GoldenMem and an interleaved training strategy to handle long-form cinematic data.

**Strengths:**

1. The paper tackles the ambitious and important challenge of long-form, multi-scene video generation, moving significantly beyond the single-shot clips that current models typically produce. This hierarchical approach of planning keyframes first and then synthesizing video is an intuitive and powerful way to maintain long-range consistency.

2. The proposed GoldenMem technique for visual context compression is a clever and efficient solution to manage the otherwise exploding computational cost of long video sequences. This method effectively preserves historical context with a bounded and minimal overhead.

**Weaknesses:**

This hierarchical style to produce narrative videos has been utilized in many previous papers, such as MovieDreamer, VideoDirectorGPT, which weakens the novelty of this paper. And the proposed Goldmem is overclaimed in my opinion.

**Questions:**

I have reviewed this paper during NIPS2026, where I rejected this paper due to the novelty. As the contributions are unchanged, I still kept my origin rating in NIPS2026.

---

### Official Review · Reviewer_2sQo · 2025-10-28

**Soundness:** 3
**Presentation:** 3
**Contribution:** 3
**Rating:** 8
**Confidence:** 4

**Summary:**

Captain Cinema proposes a two-stage framework that first performs top-down keyframe planning from a detailed movie storyline and then uses bottom-up video synthesis conditioned on those keyframes, training a Multimodal Diffusion Transformer (MM-DiT) with an interleaved strategy on a curated cinematic dataset to produce short multi-scene movies with long-range coherence. GoldenMem is the paper’s long-range memory mechanism that stores a small set of high-quality “golden” keyframe representations (visual and/or multimodal embeddings) and uses them as persistent anchors during interleaved training and inference to maintain character, scene, and stylistic consistency across many scenes and long context window.

**Strengths:**

1. Clear decomposition into planning + synthesis, separating long-range narrative/keyframe planning from local spatio-temporal generation addresses coherence at different timescales and makes the overall task more tractable.

2. Long-context modeling focus of adapting MM-DiT with an interleaved training strategy targets one of the core failure modes of generative video (losing global narrative or character consistency across scenes).

3. Use of keyframes as conditioning signals where explicit keyframes act as strong structural constraints that likely improve framing, character consistency, and scene layout compared with unconstrained long-video diffusion.

4. Building a dataset of interleaved data pairs tailored to multi-scene narrative generation helps align training data to the intended output domain and probably improves qualitative results.

5. Practical pipeline for short-movie use case, the paper presents an end-to-end system that maps a storyline to a finished short movie, which is a meaningful step toward usable creative tools for filmmakers and storytellers

6. GoldenMem keeps long-range visual anchors that directly enforce consistency for recurring characters and settings, which is a central failure mode in multi-scene movie generation. Compact and computationally efficient relative to storing entire past frames or attending over extremely long context lengths, improving scalability for multi-scene generation. It is easy to integrate into a two-stage pipeline (planner → keyframes → generator) because golden entries are naturally derived from planned keyframes and can be updated or replaced as the storyline evolves

**Weaknesses:**

1. The method’s success hinges on the generated keyframes accurately reflecting the narrative and desired visual style; failure modes or hallucinated keyframes could produce hard-to-correct artifacts in the final video. Analysis on that could be useful.

2. Conditioning on keyframes constrains endpoints, but plausible and temporally coherent transitions between them remain challenging; motion realism, continuity of lighting, and consistent character actions may still suffer. Any comments on that could be useful.

3. Cinematic quality and narrative consistency are subjective; the paper relies heavily on qualitative examples and user studies rather than standardized, reproducible metrics, making comparisons and objective evaluation harder.

4. MM-DiT long-context training with interleaved pairs and high-resolution cinematic data  requires substantial compute and curated data, raising barriers to replication and broader adoption.

**Questions:**

1. What exactly is the interleaving schedule and curriculum for MM-DiT (sequence lengths, mixing ratios, optimization schedule), and how sensitive are results to these hyperparameters?

2. Does the system include mechanisms (e.g., latent sampling, clarification prompts, fallback heuristics) when the storyline lacks detail, and how does output diversity vs. fidelity trade off in those cases?

3. Are there explicit modules or losses that enforce frame-to-frame continuity, or is it emergent from training?

4. What are the common, reproducible failure modes (e.g., broken limbs, character swapping, sudden lighting jumps), and are there proposed mitigation strategies?

5. How well does the model adapt to different cinematic genres, lighting conditions, actor types, or animation styles; is fine-tuning required per style?

6. What are the sources and licensing terms of the curated cinematic dataset, how diverse is it, and how might dataset bias affect what stories/styles the model can generate?

7. How does the system prevent or detect generation that imitates living actors or copyrighted content without consent, and are there guardrails against misuse?

---

### Official Review · Reviewer_uMzc · 2025-10-31

**Soundness:** 3
**Presentation:** 4
**Contribution:** 3
**Rating:** 8
**Confidence:** 4

**Summary:**

This work introduces Captain Cinema, a DiT-based framework for short movie generation. The key idea of this framework lies in bridging the top-down and bottom-up interleaved generation paradigms. Given the storyboard, a top-down keyframe planner first produce a sequence of key frames, then a bottom-up video generator interpolates full motion conditioned on these keyframes. In addition, a memory-efficient mechanism (i.e., GoldenMem) is proposed to address the explosive increase in token budget. Comprehensive evaluation and analysis demonstrate the satisfactory visual quality, consistency, and efficiency of the proposed method for multi-scene video generation.

**Strengths:**

1. This work makes a preliminary exploration of multi-scene, whole-movie generation. Its data acquisition and processing, as well as the proposed keyframe planning and keyframe-conditioned video generation, will contribute to the community and inspire future work.

2. The introduced GoldenMem mechanism is technically sound and effective, significantly reducing the token budget under long contexts, as shown in Figure 4. Both GoldenMem and Hybrid Attention Masking alleviate computational explosion in long-context visual generation.

3. The experiments are solid and comprehensive, including qualitative and quantitative evaluation, ablation analysis, and stress testing.

4. The paper is clear and well-structured.

**Weaknesses:**

1. There appears to be no qualitative comparison between Captain Cinema and existing methods. Based on the provided videos, the diversity and richness of the generated scenes are still limited, with no significant difference compared to LCT.

2. The GoldenMem mechanism is somewhat similar to FramePack, which diminishes its novelty.

3. The keyframe-based approach cannot guarantee temporal consistency between shots, and the proposed keyframe-conditioned video generation model does not seem to be able to alleviate the problem either.

4. Figure 1 claims that Captain Cinema can generate multi-scene videos longer than 1,000 seconds, but no demo video or evidence of this is provided.

5. L254-L255: "geneeration" -> "generation"

**Questions:**

1. I'm curious why this work uses a keyframe-based approach for multi-scene video generation, instead of a native approach like LCT?

---

### Official Review · Reviewer_c7bN · 2025-10-31

**Soundness:** 3
**Presentation:** 2
**Contribution:** 3
**Rating:** 4
**Confidence:** 3

**Summary:**

This paper introduces "Captain Cinema," a novel framework for generating short, multi-scene movies from detailed textual storylines. The core idea is a two-stage process: 1) a "top-down keyframe planning" stage, which generates a sequence of narratively coherent keyframes that act as a storyboard, and 2) a "bottom-up video synthesis" stage, which generates the video motion between these keyframes. To tackle the challenges of long-range consistency and exploding context length, the authors propose several key contributions. The most significant is "GoldenMem," a memory compression mechanism that stores past keyframes at progressively lower resolutions based on an inverse-Fibonacci sequence derived from the golden ratio, allowing the model to maintain a long visual history with a fixed token budget. The framework also incorporates a hybrid attention mechanism for Multimodal Diffusion Transformers (MM-DiT), progressive long-context finetuning, and dynamic stride sampling to enable stable and efficient training on long-form cinematic data. The authors demonstrate through extensive experiments, including a long-context stress test and detailed ablations, that Captain Cinema can generate narratively and visually consistent short movies that surpass the capabilities of existing methods.

**Strengths:**

1. The overall framework is simple and makes sense. It mainly consists of a two-stage Architecture: The decomposition of the problem into top-down planning (storyboarding) and bottom-up synthesis (animating) is logical and effective. This enables multi-scene generation with better narrative coherence.
2. GoldenMem Context Compression: The GoldenMem technique introduces an inverse-Fibonacci downsampling protocol, maintaining rich long-range context while keeping token budgets tractable. Ablative analysis convincingly demonstrates that this enables longer context windows and computational savings without major quality loss.
3. Visualization and Analysis: The figures are diverse and directly illustrate methodological claims and empirical results.

**Weaknesses:**

1. **Limited Theoretical Analysis of GoldenMem**: The mathematical treatment of GoldenMem in Section 3.2, while operationally clear, lacks a principled analysis of compression trade-offs. The geometric decay is motivated by token counts, but there is little insight into semantic loss or the optimality of the selection policy. For example, while the summation gives a neat token bound, the information preserved under this decay is not theoretically or empirically analyzed—could less aggressive or adaptive schemes outperform?
2. **Context Selection & Semantic Retrieval Ambiguity**: The approach for semantic-oriented context fetching via CLIP/T5 is described but lacks sufficient detail for reproducibility and interpretation. For example, in Figure 5, it's claimed that CLIP-based retrieval leads to better coverage—but how is "coverage" computed? There is no ablation showing the impact on downstream movie coherence or consistency.
3. **Unclear hybrid attention mechanism**: The paper provides insufficient clarity on how the hybrid attention masking is applied in training versus inference.

**Questions:**

**Major Comments**:
1. What was the rationale for choosing the golden ratio specifically, beyond the aesthetic connection to the "golden ratio squares" analogy? How sensitive is performance to the golden ratio choice? Have you tried φ variants (e.g., 1.5, 2.0) or adaptive learned scales?
2. Masking schedule: Do you train with bidirectional global attention and switch to causal at inference? Please clarify the exact masking configurations for both training and inference.
3. How is the hybrid attention masking applied differently during training and inference? Can you provide quantitative or qualitative evidence of its impact on model performance?

**Minor Comments**:
1. I agree that multi-scene movie generation is a meaningful and valuable task. The proposed approach—first generating key frames and then extending them into a video—is conceptually similar to story visualization tasks (e.g., ConsiStory, DramStory, StoryBooth). These related works deserve more thorough discussion and comparison.
2. The overall paper reads more like a technical report, providing only a brief overview of the methods. It would be more convincing if the authors could elaborate on the insights and motivations behind the proposed techniques.

I would be happy to raise my overall score if all concerns are properly addressed.

---

### Meta-Review · Area_Chair_w8iP · 2026-01-06

**Summary:**

Paper was evaluated by reviewed by four reviewers and received two positive (8) and two marginally below acceptance (4) scores. Overall, reviewers appreciated simplicity of the approach that clearly decomposes planning + synthesis as well as long-context modeling (through GoldenMem). At the same time, a number of concerns have been identified, including:

1) Limited Theoretical Analysis of GoldenMem [c7bN]
2) Lacking description of context selection & semantic retrieval ambiguity, as well as corresponding ablations. [c7bN]
3) Insufficient clarity on how the hybrid attention masking is applied (e.g., in training versus inference). [c7bN]
4) Potential lack of qualitative improvements over LCT, particularly with respect to diversity of generated scenes. [uMzc]
5) Lack of GoldenMem novelty with respect to FramePack [uMzc], and novelty in general [TXMh]
6) Lacking supporting evidence the "Captain Cinema can generate multi-scene videos longer than 1,000 seconds" [uMzc]
7) Potential issues with coherence between frames [uMzc, 2sQo]

Unfortunately, authors have failed to provide a rebuttal or revisions to the paper that address these concerns. At the same time, the approach was deemed interesting and potentially valuable. After discussion between AC and SAC, it was agreed that positives outweighed the negatives and recommendation of Acceptance was reached. At the same time, authors are VERY STRONGLY encouraged to address at least (1), (2) and (3) in the camera ready through revisions in the paper which should include additional ablations being requested.

**Reviewer Concerns:**

1) Limited Theoretical Analysis of GoldenMem [c7bN]
2) Lacking description of context selection & semantic retrieval ambiguity, as well as corresponding ablations. [c7bN]
3) Insufficient clarity on how the hybrid attention masking is applied (e.g., in training versus inference). [c7bN]
4) Potential lack of qualitative improvements over LCT, particularly with respect to diversity of generated scenes. [uMzc]
5) Lack of GoldenMem novelty with respect to FramePack [uMzc], and novelty in general [TXMh]
6) Lacking supporting evidence the "Captain Cinema can generate multi-scene videos longer than 1,000 seconds" [uMzc]
7) Potential issues with coherence between frames [uMzc, 2sQo]

**Reviewer Scores:**

No rebuttal is provided, so reviewers would not have raised the score. Further, I believe that lack of rebuttal would have likely resulted in the paper being rated lower after reviewer discussion given the collective reviewer concerns.

---

### Decision · Program_Chairs · 2026-01-26

Accept (Poster)